# Exact Certification of (Graph) Neural Networks Against Label Poisoning

**Mahalakshmi Sabanayagam**[1*],    **Lukas Gosch**[1,2,3*],    **Stephan Günnemann**[1,2,3],
**Debarghya Ghoshdastidar**[1,2]
[1] School of Computation, Information and Technology, Technical University of Munich
[2] Munich Data Science Institute [3] Munich Center for Machine Learning (MCML); Germany
`{m.sabanayagam, l.gosch, s.guennemann, d.ghoshdastidar}@tum.de`

## Abstract

Machine learning models are highly vulnerable to label flipping, i.e., the adversarial modification (poisoning) of training labels to compromise performance. Thus, deriving robustness certificates is important to guarantee that test predictions remain unaffected and to understand worst-case robustness behavior. However, for Graph Neural Networks (GNNs), the problem of certifying label flipping has so far been unsolved. We change this by introducing an *exact certification* method, deriving both *sample-wise* and *collective* certificates. Our method leverages the Neural Tangent Kernel (NTK) to capture the training dynamics of wide networks enabling us to reformulate the bilevel optimization problem representing label flipping into a Mixed-Integer Linear Program (MILP). We apply our method to certify a broad range of GNN architectures in node classification tasks. Thereby, concerning the worst-case robustness to label flipping: ($i$) we establish hierarchies of GNNs on different benchmark graphs; ($ii$) quantify the effect of architectural choices such as activations, depth and skip-connections; and surprisingly, ($iii$) *uncover a novel phenomenon* of the robustness plateauing for intermediate perturbation budgets across all investigated datasets and architectures. While we focus on GNNs, our certificates are applicable to sufficiently wide NNs in general through their NTK. Thus, our work presents the first exact certificate to a poisoning attack ever derived for neural networks, which could be of independent interest. The code is available at `https://github.com/saper0/qpcert`.

# 1 Introduction

Machine learning models are vulnerable to data poisoning where adversarial perturbations are applied to the training data to compromise the performance of a model at test time (Goldblum et al., 2023). In addition, data poisoning has been observed in practice and is recognized as a critical concern for practitioners and enterprises (Kumar et al., 2020; Grosse et al., 2023; Cinà et al., 2024). The practical feasibility was impressively demonstrated by Carlini et al. (2024), who showed that with only $60 USD they could have poisoned several commonly used web-scale datasets.

Label flipping is a special type of data poisoning where a fraction of the training labels are corrupted, leaving the features unaffected. This type of attack has proven widespread effectivity ranging from classical methods for i.i.d. or graph data (Biggio et al., 2011; Liu et al., 2019), to modern deep learning systems for images, text, or graph-based learning (Jha et al., 2023; Wan et al., 2023; Lingam et al., 2024). Exemplary, Lingam et al. (2024) showed that one adversarial label flip could reduce the accuracy of Graph Convolution Networks (GCNs) (Kipf & Welling, 2017) by over 17% on a smaller version of Cora-ML (McCallum et al., 2000). Similarly, Fig. 1a demonstrates for the Karate Club network (Zachary, 1977) that one label-flip can reduce the accuracy of a GCN by 50%.

Although several empirical defenses have been developed to counter label flipping attacks (Zhang et al., 2020; Paudice et al., 2019), they remain vulnerable to increasingly sophisticated attacks (Koh et al., 2022). This highlights the need for *robustness certificates* which offer formal guarantees

---

*Equal contribution.

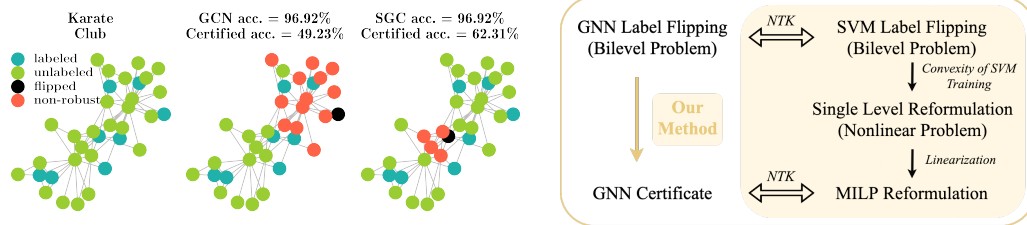

(a) Worst-case robustness to one label flip of two GNNs.  (b) Illustration of our label-flipping certificate.

Figure 1: (a) The Karate Club network is visualized with its labeled (●) and unlabeled (●) nodes. The adversarial label flip (●) calculated by our method outlined in (b) provably leads to most node predictions being flipped (●) for two GNNs (GCN & SGC). The certified accuracy refers to the percentage of correctly classified nodes that remain robust to the attack.

that the test predictions remain unaffected under a given perturbation model. However, there are currently *no works* on certifying label poisoning for Graph Neural Networks (GNNs) and as a result, little is known about the worst-case (adversarial) robustness of different architectural choices. That a difference in behavior can be expected is motivated in Fig. 1a, where exchanging the ReLU in a GCN with an identity function forming a Simplified Graph Convolutional Network (SGC) (Wu et al., 2019), results in significantly higher worst-case robustness to label flipping for Karate Club.

In general, robustness certificates can be divided into being *exact* (also known as *complete*), i.e., returning the exact adversarial robustness of a model representing its worst-case robustness to a given perturbation model, or *incomplete*, representing an underestimation of the exact robustness. Complete certificates allow us to characterize and compare the effect different architectural choices have on worst-case robustness as exemplified in Fig. 1a, whereas incomplete certificates suffer from having variable tightness for different models, making meaningful comparisons difficult (Li et al., 2023). Currently, even for i.i.d. data there are no exact poisoning certificates for NNs, and existing approaches to certify label-flipping are limited to randomized smoothing (Rosenfeld et al., 2020) and partition-based aggregation (Levine & Feizi, 2021), which offer incomplete guarantees for smoothed or ensembles of classifiers. Thus, adapting these techniques to graphs will not enable us to understand the effect of specific architectural choices for GNNs on their worst-case robustness. The lack of exact certificates can be understood due to the inherent complexity of capturing the effect a change in the training data has on the training dynamics and consequently, is an unsolved problem. This raises the question: *is it even possible to compute exact certificates against label poisoning?*

In this work, we resolve this question by first deriving *exact sample-wise* robustness certificates for sufficiently wide NNs against label flipping and evaluate them for different GNNs focusing on semi-supervised node classification. Based on our sample-wise derivation we develop an *exact collective* certification strategy that certifies the entire test set simultaneously. This is of particular importance for poisoning certificates, as a model is usually trained once on a given training set and then evaluated. Consequently, an attacker can only choose one perturbation to the training set targeting the performance on all test points. To capture the effect of label flipping on the training process of a network, our approach takes inspiration from Gosch et al. (2024) and makes use of the Neural Tangent Kernel (NTK) of different GNNs (Sabanayagam et al., 2023), which precisely characterizes the training dynamics of sufficiently wide NNs (Arora et al., 2019). Concretely, we leverage the equivalence of a wide NN trained using the soft-margin loss with a Support Vector Machine (SVM) that uses the NTK of the network as its kernel (Chen et al., 2021). This allows us to reformulate the bilevel optimization problem describing label flipping as a Mixed-Integer Linear Program (MILP) yielding a certificate for wide NNs as illustrated in Fig. 1b. As the MILP scales with the number of labeled data, our method is a good fit to certify GNNs for semi-supervised node-classification on graphs, due to the usually encountered sparse labeling. While Gosch et al. (2024) were the first to use the NTK to derive model-specific poisoning certificates, their work is limited to feature perturbations and incomplete sample-wise certification. Thus, our contributions are:

**(i)** We derive the *first exact robustness certificates* for NNs against label flipping. Next to *sample-wise certificates* (Sec. 3.1), we develop *exact collective certificates* (Sec. 3.2) particularly important for characterizing the worst-case robustness of different architectures to label poisoning. Concretely, our certificates apply to infinite-width NNs and hold with high probability for wide finite-width NNs.

**(ii)** We apply our certificates to a wide-range of GNNs for node-classification on both, real and synthetic data (Sec. 4). Thereby, we establish that worst-case robustness hierarchies are highly

data-dependent, and quantify the effect of different graph properties and architectural choices (e.g., activations, depth, skip-connections) on worst-case robustness.

**(iii)** Using the collective certificate, we uncover a surprising phenomenon: across all datasets, most architectures show a worst-case robustness plateaus for intermediate attack budgets so far not observed with adversarial attacks (Lingam et al., 2024).

**(iv)** Beyond (wide) NNs, our MILP reformulation is valid for SVMs with arbitrary kernel choices. Thus, it is the first certificate for kernelized SVMs against label flipping.

**Notation.** We use bold upper and lowercase letters to denote a matrix $\mathbf{A}$ and vector $\mathbf{a}$, respectively. The $i$-th entry of a vector $\mathbf{a}$ is denoted by $a_i$, and the $ij$-entry of a matrix $\mathbf{A}$ by $A_{ij}$. We use the floor operator $\lfloor n \rfloor$ for the greatest integer $\leq n$, and $[n]$ to denote $\{1, 2, \ldots, n\}$. Further, $\langle ., . \rangle$ for scalar product, $\mathbb{E}[\cdot]$ for the expectation and $\mathbb{1}[.]$ for the indicator function. We use $\|\cdot\|_p$ with $p = 2$ for vector Euclidean norm and matrix Frobenius norm, and $p = 0$ for vector 0-norm.

## 2 PROBLEM SETUP AND PRELIMINARIES

We consider semi-supervised node classification, where the input graph $\mathcal{G} = (\boldsymbol{S}, \boldsymbol{X})$ contains $n$ nodes, each associated with a feature vector $\boldsymbol{x}_i \in \mathbb{R}^d$ aggregated in the feature matrix $\boldsymbol{X} \in \mathbb{R}^{n \times d}$. Graph structure is encoded in $\boldsymbol{S} \in \mathbb{R}_{\geq 0}^{n \times n}$, typically representing a type of adjacency matrix. Labels $\mathbf{y} \in \{1, \ldots, K\}^m$ are provided for a subset of $m$ nodes ($m \leq n$). Without loss of generality, we assume that the first $m$ nodes are labeled. The objective is to predict the labels for the $n - m$ unlabeled nodes in a transductive setting or to classify newly added nodes in an inductive setting.

**GNNs.** An $L$-layer GNN $f_\theta$ with learnable parameters $\theta$ takes the graph $\mathcal{G}$ as input and outputs a prediction for each node with $f_\theta(\mathcal{G}) \in \mathbb{R}^{n \times K}$ for multiclass and $f_\theta(\mathcal{G}) \in \mathbb{R}^n$ for binary classification; the output for a node $i$ is denoted by $f_\theta(\mathcal{G})_i$. We consider GNNs with a linear output layer parameterized using weights $\boldsymbol{W}^{(L+1)}$ and refer to Sec. 4 for details on the used architectures.

**Infinite-width GNNs and the Neural Tangent Kernel.** When the width of $f_\theta$ goes to infinity and the parameters are initialized from a Gaussian $\mathcal{N}(0, 1/\text{width})$, the training dynamics of $f_\theta$ are exactly described by its NTK (Jacot et al., 2018; Arora et al., 2019). For node classification, the NTK of a model $f_\theta$ is defined between two nodes $i$ and $j$ as $Q_{ij} = \mathbb{E}_\theta[\langle \nabla_\theta f_\theta(\mathcal{G})_i, \nabla_\theta f_\theta(\mathcal{G})_j \rangle]$ (Sabanayagam et al., 2023), where the expectation is taken over the parameter initialization.

**On the Equivalence to Support Vector Machines.** In the following, we focus on binary node classification with $y_i \in \{\pm 1\}$ and refer to App. A for the multi-class case. We learn the parameters $\theta$ of a GNN by optimizing the soft-margin loss by gradient descent:

$$\min_\theta \mathcal{L}(\theta, \mathbf{y}) = \min_\theta \sum_{i=1}^{m} \max(0, 1 - y_i f_\theta(\mathcal{G})_i) + \frac{1}{2C} \|\mathbf{W}^{(L+1)}\|_2^2 \tag{1}$$

where $C > 0$ is a regularization constant. In the infinite-width limit, the training dynamics for Eq. (1) are the same as those of an SVM with $f_\theta$'s NTK as kernel. Thus, solving Eq. (1) is equivalent to solving the dual problem of an SVM without bias (Gosch et al., 2024; Chen et al., 2021):

$$\mathrm{P}_1(\mathbf{y}): \min_{\boldsymbol{\alpha}} -\sum_{i=1}^{m} \alpha_i + \frac{1}{2} \sum_{i=1}^{m} \sum_{j=1}^{m} y_i y_j \alpha_i \alpha_j Q_{ij} \text{ s.t. } 0 \leq \alpha_i \leq C \; \forall i \in [m] \tag{2}$$

where $\boldsymbol{\alpha} \in \mathbb{R}^m$ are the SVM dual variables, and $Q_{ij}$ the NTK of $f_\theta$ between nodes $i$ and $j$. The solution to Eq. (2) is not guaranteed to be unique; hence, denote by $\mathcal{S}(\mathbf{y})$ the set of $\boldsymbol{\alpha}$ vectors solving $\mathrm{P}_1(\mathbf{y})$. Given any $\boldsymbol{\alpha}$, an SVM predicts the label of a node $t$ by computing $\text{sign}(\sum_{i=1}^{m} y_i \alpha_i Q_{ti})$.

**On Finite-width GNN Certification using NTK.** Any exact certificate derived for SVM with NTK as its kernel directly provides exact deterministic guarantees for infinite-width GNNs through their equivalence. Concerning the finite-width case, where $w$ denotes the smallest layer-width of the GNN, the output difference to the SVM is bounded by $\mathcal{O}(\frac{\ln w}{\sqrt{w}})$ with probability $1 - \exp(-\Omega(w))$ as shown in Gosch et al. (2024); Liu et al. (2020) (see App. E for more model-specific guarantees). Thus, for increasing $w$ the output difference approaches 0 while the probability approaches 1. Note that the certificate becomes incomplete for a fixed finite but not sufficiently wide network.

**Label Poisoning.** We assume that before training the adversary $\mathcal{A}$ has control over the labels of an $\epsilon$-fraction of labeled nodes. Formally, $\mathcal{A}$ can choose perturbed labels $\tilde{\mathbf{y}} \in \mathcal{A}(\mathbf{y}) := \{\tilde{\mathbf{y}} \in [K]^m \mid \|\tilde{\mathbf{y}} - \mathbf{y}\|_0 \leq \lfloor \epsilon m \rfloor\}$ with the goal to minimize the correct predictions of test nodes as described by an attack objective $\mathcal{L}_{att}(\theta, \tilde{\mathbf{y}})$ after training on $\tilde{\mathbf{y}}$. This can be written as a bilevel optimization problem

$$\min_{\theta, \tilde{\mathbf{y}}} \mathcal{L}_{att}(\theta, \tilde{\mathbf{y}}) \quad \text{s.t.} \quad \tilde{\mathbf{y}} \in \mathcal{A}(\mathbf{y}) \ \wedge \ \theta \in \arg\min_{\theta'} \mathcal{L}(\theta', \tilde{\mathbf{y}}). \tag{3}$$

**Prior Work on Poisoning and its Bilevel Formulation.** Developing poisoning *attacks* by approximately solving the associated bilevel problem is common for SVMs (Biggio et al., 2012), deep networks (Muñoz-González et al., 2017; Koh et al., 2022), and GNNs alike (Zügner & Günnemann, 2019). From these, we highlight Mei & Zhu (2015) who focus on SVMs and similar to us, transform the bilevel problem into a single-level one, but only approximately solve it with a gradient-based approach and don't consider label flipping. Regarding label flipping, Biggio et al. (2011) and Xiao et al. (2012) develop attacks for SVMs solving Eq. (3) with non-gradient based heuristics; Lingam et al. (2024) create an attack for GNNs by solving Eq. (3) with a regression loss, replacing the GCN with a surrogate model given by the NTK. Concerning *certificates* for data poisoning, there are only few works with none providing exact guarantees. The approaches based on differential privacy (Ma et al., 2019), randomized smoothing (Rosenfeld et al., 2020; Lai et al., 2024), and majority voting (Levine & Feizi, 2021) are inherently incomplete. In contrast, similar to us, Gosch et al. (2024) directly solve the bilevel formulation to obtain sample-wise feature poisoning certificates for wide (G)NNs. However, their reformulation it not exact or applicable to the label flipping problem, and they do not provide a collective certificate. We detail the technical differences in App. F.

## 3 LABELCERT FOR LABEL POISONING

Our derivation of label flipping certificates for GNNs is fundamentally based on the equivalence with an SVM using the NTK of the corresponding network as its kernel. Concretely, our derivations follow three high-level steps depicted in Fig. 1b: $(i)$ we instantiate the bilevel problem in Eq. (3) for (kernelized) SVMs with a loss describing misclassification and using properties of the SVM's dual formulation, we transform it into a single-level non-linear optimization problem; $(ii)$ we introduce linearizations of the non-linear terms, allowing us to further reformulate the non-linear problem into an equivalent mixed-integer linear program; and $(iii)$ by choosing the NTK of a network as the kernel, solving the resulting MILP yields a certificate for the corresponding sufficiently-wide NN. In Sec. 3.1 we present our sample-wise certificate for label flipping and then, derive a collective certification strategy in Sec. 3.2. We note that the reformulation process requires no approximations or relaxations; hence, the derived certificates are exact. In what follows, we choose an SVM in its dual formulation as our model, hence the model parameters $\theta$ are the dual variables $\boldsymbol{\alpha}$. Further, we present the certificates for binary labels $y_i \in \{\pm 1\}$ and discuss the multi-class case in App. A.

### 3.1 SAMPLE-WISE CERTIFICATION

To obtain a sample-wise certificate, we have to prove that the model prediction for a test node $t$ can't be changed by training on any $\tilde{\mathbf{y}} \in \mathcal{A}(\mathbf{y})$. Let $\boldsymbol{\alpha}^*$ be an optimal solution to the dual problem $P_1(\mathbf{y})$ obtained by training on the original labels $\mathbf{y}$ and denote by $\hat{p}_t = \sum_{i=1}^m y_i \alpha_i^* Q_{ti}$ the corresponding SVM's prediction for $t$. Similary, let $\boldsymbol{\alpha}$ be an optimal solution to $P_1(\tilde{\mathbf{y}})$ with perturbed labels $\tilde{\mathbf{y}}$ and the new prediction be $p_t = \sum_{i=1}^m \tilde{y}_i \alpha_i Q_{ti}$. As an SVM assigns class based on the sign of its prediction, the class prediction changes if and only if $\text{sign}(\hat{p}_t) \cdot p_t < 0$ [1]. Thus, the bilevel problem

$$P_2(\mathbf{y}) : \min_{\boldsymbol{\alpha}, \tilde{\mathbf{y}}} \text{sign}(\hat{p}_t) \sum_{i=1}^m \tilde{y}_i \alpha_i Q_{ti} \quad s.t. \quad \tilde{\mathbf{y}} \in \mathcal{A}(\mathbf{y}) \ \wedge \ \boldsymbol{\alpha} \in \mathcal{S}(\tilde{\mathbf{y}}) \tag{4}$$

certifies robustness, if the optimal solution is $> 0$. However, bilevel problems are notoriously hard to solve (Schmidt & Beck, 2023), making $P_2(\mathbf{y})$ intractable in its current form. Now, notice that the inner optimization problem $\boldsymbol{\alpha} \in \mathcal{S}(\tilde{\mathbf{y}})$ consists of the SVM's dual problem $P_1(\tilde{\mathbf{y}})$, which is convex and fulfills Slater's condition for every $\tilde{\mathbf{y}}$ (see App. B). Thus, we can replace $\boldsymbol{\alpha} \in \mathcal{S}(\tilde{\mathbf{y}})$ with $P_1(\tilde{\mathbf{y}})$'s Karush-Kuhn-Tucker (KKT) conditions to obtain **a single-level problem** $P_3(\mathbf{y})$ that shares

---

[1]In our implementation, we treat the undefined case of $\hat{p}_t \cdot p_t = 0$ as misclassification.

the same optimal solutions as $P_2(\mathbf{y})$ (Dempe & Dutta, 2012). The KKT conditions define three sets of constraints. First, stationarity constraints from the derivate of the Lagrangian of $P_1(\tilde{\mathbf{y}})$:

$$\forall i \in [m] : \quad \sum_{j=1}^{m} \tilde{y}_i \tilde{y}_j \alpha_j Q_{ij} - 1 - u_i + v_i = 0 \tag{5}$$

where $\mathbf{u}, \mathbf{v} \in \mathbb{R}^m$ are the Lagrangian dual variables. Secondly, feasibility ranges for all $i \in [m]$: $\alpha_i \geq 0$, $C - \alpha_i \geq 0$, $u_i \geq 0$, $v_i \geq 0$, and lastly, the complementary slackness constraints:

$$\forall i \in [m] : \quad u_i \alpha_i = 0, \ v_i(C - \alpha_i) = 0. \tag{6}$$

Thus, the resulting single-level optimization problem $P_3(\mathbf{y})$ now optimizes over $\boldsymbol{\alpha}, \tilde{\mathbf{y}}, \mathbf{u}$ and $\mathbf{v}$.

**A Mixed-Integer Linear Reformulation.** $P_3(\mathbf{y})$ is a difficult to solve non-linear problem as Eq. (5) defines multilinear constraints and both, the objective and Eq. (6) are bilinear. Thus, to make $P_3(\mathbf{y})$ tractable, we introduce (exact) linearizations of all non-linearities, as well as linearly model the adversary $\tilde{\mathbf{y}} \in \mathcal{A}(\mathbf{y})$.

($i$) *Modeling the adversary:* First, we have to ensure that the variable $\tilde{\mathbf{y}} \in \{-1, 1\}^m$. To do so, we model $\tilde{\mathbf{y}}$ as being continuous and introduce a binary variable $\mathbf{y}' \in \{0, 1\}^m$ that enforces $\tilde{\mathbf{y}} \in \{-1, 1\}^m$ through adding the constraint $\tilde{y}_i = 2y_i' - 1$ for all $i \in [m]$. Then, the bounded perturbation strength $\|\tilde{\mathbf{y}} - \mathbf{y}\|_0 \leq \lfloor \epsilon m \rfloor$ can be formulated as:

$$\sum_{i=1}^{m} 1 - y_i \tilde{y}_i \leq 2 \lfloor \epsilon m \rfloor. \tag{7}$$

($ii$) *Objective and Stationarity constraint:* The non-linear product terms in the objective can be linearized by introducing a new variable $\mathbf{z} \in \mathbb{R}^m$ with $z_i = \alpha_i \tilde{y}_i$. Since for all $i \in [m]$ it holds that $0 \leq \alpha_i \leq C$ and $\tilde{y}_i \in \{\pm 1\}$, the multiplication $z_i = \alpha_i \tilde{y}_i$ can be modeled by

$$\forall i \in [m] : \quad -\alpha_i \leq z_i \leq \alpha_i, \ \alpha_i - C(1 - \tilde{y}_i) \leq z_i \leq C(1 + \tilde{y}_i) - \alpha_i. \tag{8}$$

Thus, replacing all product terms $\alpha_i \tilde{y}_i$ in $P_3(\mathbf{y})$ with $z_i$ and adding the linear constraints of Eq. (8) resolves the non-linearity in the objective. As the product terms also appear in the stationarity constraints of Eq. (5), they become bilinear reading $\forall i \in [m]$, $\sum_{j=1}^{m} \tilde{y}_i z_j Q_{ij} - 1 - u_i + v_i = 0$. As the non-linear product terms $\tilde{y}_i z_j$ in the stationarity constraints are also the multiplication of a binary with a continuous variable, we linearize them following a similar strategy. We introduce a new variable $\mathbf{R} \in \mathbb{R}^{m \times m}$ with $R_{ij}$ representing $\tilde{y}_i z_j$ and replace all occurrences of $\tilde{y}_i z_j$ with $R_{ij}$. Then, as $-C \leq z_j \leq C$ we model $R_{ij} = \tilde{y}_i z_j$ by adding the linear constraints

$$\forall i, j \in [m] : -C(1 + \tilde{y}_i) \leq R_{ij} + z_j \leq C(1 + \tilde{y}_i), \ -C(1 - \tilde{y}_i) \leq R_{ij} - z_j \leq C(1 - \tilde{y}_i) \tag{9}$$

resolving the remaining non-linearity in the stationarity constraint.

($iii$) *Complementary Slackness constraints:* The bilinear complementary slackness constraints in Eq. (6) represent conditionals: *if $\alpha_i > 0$ then $u_i = 0$ else $u_i \geq 0$* and similar for $v_i$. Thus, we model them using an equivalent big-M formulation:

$$\forall i \in [m] : \quad u_i \leq M_{u_i} s_i, \ \alpha_i \leq C(1 - s_i), \ s_i \in \{0, 1\},$$
$$v_i \leq M_{v_i} t_i, \ C - \alpha_i \leq C(1 - t_i), \ t_i \in \{0, 1\}. \tag{10}$$

where we introduce new binary variables $\mathbf{s}, \mathbf{t} \in \{0, 1\}^m$ and large positive constants $M_{u_i}$ and $M_{v_i}$ for each $i \in [m]$. Usually, defining valid big-M's for complementary slackness constraints is prohibitively difficult (Kleinert et al., 2020). However, in App. C we show how to use special structure in our problem to set valid and small big-M values, not cutting away the relevant optimal solutions to $P_3(\mathbf{y})$.

With all non-linear terms in $P_3(\mathbf{y})$ linearized and having modeled $\tilde{\mathbf{y}} \in \mathcal{A}(\mathbf{y})$, we can now state:

**Theorem 1 (Sample-wise MILP)** *Given the adversary $\mathcal{A}$ and positive constants $M_{u_i}$ and $M_{v_i}$ set as in App. C for all $i \in [m]$, the prediction for node $t$ is certifiably robust if the optimal solution to*

*the MILP* $P(\mathbf{y})$*, given below, is greater than zero and non-robust otherwise.*

$$P(\mathbf{y}): \min_{\substack{\boldsymbol{\alpha},\tilde{\mathbf{y}},\mathbf{y}',\mathbf{z} \\ \mathbf{u},\mathbf{v},\mathbf{s},\mathbf{t},\mathbf{R}}} \operatorname{sign}(\hat{p}_t) \sum_{i=1}^{m} z_i Q_{ti} \quad s.t. \quad \sum_{i=1}^{m} 1 - y_i \tilde{y}_i \leq 2\lfloor \epsilon m \rfloor, \; \forall i \in [m]: \; \tilde{y}_i = 2y_i' - 1$$

$$\forall i,j \in [m]: \sum_{j=1}^{m} R_{ij} Q_{ij} - 1 - u_i + v_i = 0, \quad u_i \geq 0, \quad v_i \geq 0, \quad y_i' \in \{0,1\},$$
$$-C(1 + \tilde{y}_i) \leq R_{ij} + z_j \leq C(1 + \tilde{y}_i), \quad -C(1 - \tilde{y}_i) \leq R_{ij} - z_j \leq C(1 - \tilde{y}_i),$$
$$-\alpha_i \leq z_i \leq \alpha_i, \quad \alpha_i - C(1 - \tilde{y}_i) \leq z_i \leq C(1 + \tilde{y}_i) - \alpha_i,$$
$$u_i \leq M_{u_i} s_i, \quad \alpha_i \leq C(1 - s_i), \quad v_i \leq M_{v_i} t_i, \quad \alpha_i \geq C t_i, \quad s_i \in \{0,1\}, \quad t_i \in \{0,1\}.$$

**Computational Complexity.** The inputs to MILP $P(\mathbf{y})$ are computed in polynomial time: the NTK $Q$ in $\mathcal{O}(m^2)$ and the positive constants $M_u$ and $M_v$ in $\mathcal{O}(m)$. While these contribute polynomial complexity, the overall computation of the certificate is dominated by the MILP solution process, which is NP-hard with exponential complexity. Thus, the computation is dominated by the MILP whose runtime strongly correlates with the number of integer variables. $P(\mathbf{y})$ has in total $3m$ binary variables and thus, it gets more difficult to solve as the number of labeled training data increases.

## 3.2 COLLECTIVE CERTIFICATION

For collective certification, the objective is to compute the number of test predictions that are simultaneously robust to any $\tilde{\mathbf{y}} \in \mathcal{A}(\mathbf{y})$. This implies that the adversary is restricted to choose only one $\tilde{\mathbf{y}}$ to misclassify a maximum number of nodes. Thus, it is fundamentally different from sample-wise certification, which certifies each test node independently. Let $\mathcal{T}$ be the set of test nodes. Then, the collective certificate can be formulated using Eq. (3) by choosing to maximize $\sum_{t \in \mathcal{T}} \mathbb{1}[\hat{p}_t \neq p_t]$ as:

$$C_1(\mathbf{y}): \max_{\boldsymbol{\alpha},\tilde{\mathbf{y}}} \sum_{t \in \mathcal{T}} \mathbb{1}[\hat{p}_t \neq p_t] \quad s.t. \quad \tilde{\mathbf{y}} \in \mathcal{A}(\mathbf{y}) \wedge \boldsymbol{\alpha} \in \mathcal{S}(\tilde{\mathbf{y}}). \tag{11}$$

Following the sample-wise certificate, we transform the bilevel problem $C_1(\mathbf{y})$ into a single-level one, by replacing the inner problem $\boldsymbol{\alpha} \in \mathcal{S}(\tilde{\mathbf{y}})$ with its KKT conditions. Then, we apply the same linear modeling techniques for the stationarity and complementary slackness constraints, as well as for the adversary. To tackle the remaining non-linear objective, we first introduce a new variable $\mathbf{c} \in \{0,1\}^{|\mathcal{T}|}$ where $c_t = \mathbb{1}[\hat{p}_t \neq p_t] \; \forall t \in \mathcal{T}$ and write the single-level problem obtained so far as:

$$C_2(\mathbf{y}): \max_{\substack{\mathbf{c},\boldsymbol{\alpha},\tilde{\mathbf{y}},\mathbf{y}',\mathbf{z} \\ \mathbf{u},\mathbf{v},\mathbf{s},\mathbf{t},\mathbf{R}}} \sum_{t \in \mathcal{T}} c_t \quad s.t. \quad p_t = \sum_{i=1}^{m} z_i Q_{ti}, \text{ constraints of } P(\mathbf{y}),$$
$$\forall t \in \mathcal{T}: \text{ if } \operatorname{sign}(\hat{p}_t) \cdot p_t > 0 \text{ then } c_t = 0 \text{ else } c_t = 1.$$

Now, notice that because $-C \leq z_i \leq C$ for all $i \in [m]$, $p_t$ is bounded as $-C \sum_{i=1}^{m} |Q_{ti}| \leq p_t \leq C \sum_{i=1}^{m} |Q_{ti}|$ for all $t \in \mathcal{T}$. Let $l_t$ and $h_t$ be the respective lower and upper bounds to $p_t$. Then, we can linearize the conditional constraints in $C_2(\mathbf{y})$:

$$\forall t \in \mathcal{T}: \forall \hat{p}_t > 0: p_t \leq h_t(1 - c_t), \; p_t \geq l_t c_t, \quad \forall \hat{p}_t < 0: p_t \geq l_t(1 - c_t), \; p_t \leq h_t c_t. \tag{12}$$

As a result, we can state the following theorem (in App. D we formally write out all constraints):

**Theorem 2 (Collective MILP)** *Given the adversary* $\mathcal{A}$*, positive constants* $M_{u_i}$ *and* $M_{v_i}$ *set as in App. C for all* $i \in [m]$*, and* $\mathbf{l}, \mathbf{h} \in \mathbb{R}^{|\mathcal{T}|}$ *with* $l_t = -C \sum_{i=1}^{m} |Q_{ti}|$ *and* $h_t = C \sum_{i=1}^{m} |Q_{ti}|$*, the maximum number of test nodes that are certifiably non-robust is given by the MILP* $C(\mathbf{y})$*.*

$$C(\mathbf{y}): \max_{\substack{\mathbf{c},\boldsymbol{\alpha},\tilde{\mathbf{y}},\mathbf{y}',\mathbf{z} \\ \mathbf{u},\mathbf{v},\mathbf{s},\mathbf{t},\mathbf{R}}} \sum_{t \in \mathcal{T}} c_t \quad s.t. \quad \textit{constraints of } P(\mathbf{y}), \; \forall t \in \mathcal{T}: p_t = \sum_{i=1}^{m} z_i Q_{ti}, \; c_t = \{0,1\},$$
$$\forall t \in \mathcal{T}: \forall \hat{p}_t > 0: p_t \leq h_t(1 - c_t), \; p_t \geq l_t c_t, \quad \forall \hat{p}_t < 0: p_t \geq l_t(1 - c_t), \; p_t \leq h_t c_t.$$

**Computational Complexity.** $C(\mathbf{y})$ has $3m + |\mathcal{T}|$ binary variables. Thus, the larger the set to verify, the more complex to solve the MILP.

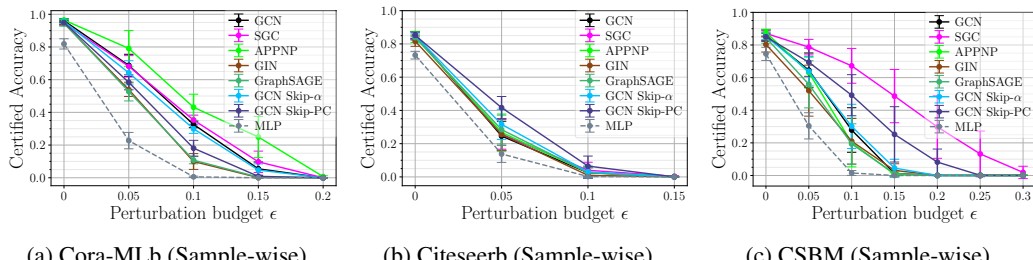

(a) Cora-MLb (Sample-wise)   (b) Citeseerb (Sample-wise)   (c) CSBM (Sample-wise)

Figure 2: Certified accuracies as given by our sample-wise certificate, for multi-class Cora-ML and Citeseer see App. I and other datasets in App. H.1. A clear and consistent hierarchy emerges across perturbation budgets concerning the worst-case robustness of different GNNs.

## 4   EXPERIMENTAL RESULTS

In Sec. 4.1 we thoroughly investigate our sample-wise and collective certificates. Sec. 4.2 discusses in detail the effect of architectural choices and graph structure.

**Datasets.** We use the real-world graph datasets Cora-ML (Bojchevski & Günnemann, 2018) and Citeseer (Giles et al., 1998) for multi-class certification. We evaluate binary class certification using *Polblogs* (Adamic & Glance, 2005), and by extracting the subgraphs containing the top two largest classes from Cora-ML, Citeseer, Wiki-CS (Mernyei & Cangea, 2020), Cora (McCallum et al., 2000) and Chameleon (Rozemberczki et al., 2021) referring to these as *Cora-MLb*, *Citeseerb*, *Wiki-CSb*, *Corab* and *Chameleonb*, respectively. To investigate the influence of graph-specific properties on the worst-case robustness, we generate synthetic datasets using random graph models, the Contextual Stochastic Block Model (CSBM) (Deshpande et al., 2018) and the Contextual Barabási–Albert Model (CBA) (Gosch et al., 2023). We sample a graph of size $n = 200$ from CSBM and CBA. We refer to App. G for the sampling scheme and dataset statistics. We choose 10 nodes per class for training for all datasets, except for Citeseer, for which we choose 20. No separate validation set is needed as we perform 4-fold cross-validation (CV) for hyperparameter tuning. All results are averaged over 5 seeds (multiclass datasets: 3 seeds) and reported with their standard deviation.

**GNN Architectures.** We evaluate a broad range of convolution-based and PageRank-based GNNs: GCN (Kipf & Welling, 2017), SGC (Wu et al., 2019), GraphSAGE (Hamilton et al., 2017), GIN (Xu et al., 2019), APPNP (Gasteiger et al., 2019), and GCN with two skip-connection variants namely GCN Skip-PC and GCN Skip-$\alpha$ (Sabanayagam et al., 2023). We also evaluate an MLP. All results concern the infinite-width limit and are obtained by solving the MILPs in Thm. 1 and 2 using Gurobi 11.0.1 (Gurobi Optimization, LLC, 2023) and the GNN's NTK as derived in Gosch et al. (2024) and Sabanayagam et al. (2023). We investigate $L = \{1, 2, 4\}$ hidden layers, if not explicitly stated, $L = 1$. All other hyperparameters are chosen based on 4-fold CV, given in App. G.2. We define the row and symmetric normalizations as $\mathbf{S}_{\text{row}} = \widehat{\mathbf{D}}^{-1}\widehat{\mathbf{A}}$, $\mathbf{S}_{\text{sym}} = \widehat{\mathbf{D}}^{-1/2}\widehat{\mathbf{A}}\widehat{\mathbf{D}}^{-1/2}$ with $\widehat{\mathbf{D}}$ and $\widehat{\mathbf{A}}$ the degree and adjacency matrices of the given graph $\mathcal{G}$ with an added self-loop.

**Evaluation.** We consider perturbation budgets $\epsilon = \{0.05, 0.1, 0.15, 0.2, 0.25, 0.3, 0.5, 1\}$ for the adversary $\mathcal{A}$, and define $\mathcal{A}$'s strength as 'weak' if $\epsilon \in (0, 0.1]$, 'intermediate' if $\epsilon \in (0.1, 0.3]$ and 'strong' if $\epsilon \in (0.3, 1]$. The test set for collective certificates consists of all unlabeled nodes on CSBM and CBA, and random samples of 50 unlabeled nodes for real-world graphs. The sample-wise certificate is calculated on all unlabeled nodes. We report *certified ratios*, referring to the percentage of test-node predictions that are provably robust to $\mathcal{A}$. For sample-wise certificates, we also report *certified accuracy*, that is the percentage of correctly classified nodes that are also provably robust to $\mathcal{A}$. As our results are obtained with exact certificates, they establish a *hierarchy* of the investigated models regarding worst-case robustness to label flipping for a given dataset and $\epsilon$, which we refer to as 'robustness hierarchy' or 'robustness ranking'. Since no prior work on exact certification for label flipping exists, the only baseline for comparison is an exhaustive enumeration of all possible perturbations — infeasible for anything beyond one or two label flips.

### 4.1   CERTIFIABLE ROBUSTNESS OF GNNS TO LABEL POISONING

We start by demonstrating the effectiveness of our **sample-wise certificate** to certify a large spectrum of GNNs against label flipping on different datasets in Fig. 2. Interestingly, our certificate

Table 1: Certified ratios in [%] calculated with our exact collective certificate on different datasets for $\epsilon \in \{0.05, 0.1, 0.15\}$ (see App. H.2.1 for all $\epsilon$). As a baseline for comparison, the certified ratio of a GCN is reported. Then, for the other models, we report their *absolute change* in certified ratio compared to a GCN, i.e., their certified ratio minus the mean certified ratio of a GCN. The most robust model for a choice of $\epsilon$ is highlighted in bold, the least robust in red.

| | **Cora-MLb** | | | **Citeseerb** | | | **CSBM** | | |
|---|---|---|---|---|---|---|---|---|---|
| $\epsilon$ | 0.05 | 0.10 | 0.15 | 0.05 | 0.10 | 0.15 | 0.05 | 0.10 | 0.15 |
| **GCN** | 86.4 ± 6.1 | 55.6 ± 10.7 | 46.8 ± 6.0 | 65.6 ± 12.5 | 50.0 ± 9.7 | 41.6 ± 7.1 | 85.7 ± 6.5 | 67.0 ± 9.7 | 48.0 ± 6.0 |
| **SGC** | +2.4 ± 5.2 | **+7.6 ± 9.2** | **+2.8 ± 6.1** | +3.6 ± 9.9 | +0.4 ± 10.8 | -0.4 ± 10.5 | **+7.8 ± 2.8** | **+21.9 ± 5.0** | **+34.3 ± 8.3** |
| **APPNP** | **+3.2 ± 7.9** | +6.8 ± 12.2 | -1.2 ± 2.7 | +4.0 ± 4.5 | +0.8 ± 5.9 | -2.4 ± 7.1 | -2.1 ± 7.4 | -6.2 ± 7.2 | -1.7 ± 3.4 |
| **GIN** | -4.8 ± 4.1 | +6.0 ± 4.5 | -2.0 ± 5.5 | +6.8 ± 6.6 | +0.8 ± 8.6 | +1.2 ± 6.1 | -3.4 ± 8.7 | -2.8 ± 13.0 | +2.6 ± 9.0 |
| **GraphSAGE** | -6.0 ± 6.5 | +1.2 ± 5.2 | +1.6 ± 6.1 | +9.6 ± 5.6 | +5.2 ± 5.3 | +2.4 ± 5.2 | -0.2 ± 4.7 | -2.3 ± 8.0 | +0.6 ± 4.4 |
| **GCN Skip-$\alpha$** | -1.2 ± 6.4 | +1.6 ± 10.2 | +0.8 ± 5.6 | +10.0 ± 6.5 | +3.2 ± 7.0 | +2.0 ± 5.4 | -0.1 ± 6.4 | -0.9 ± 8.0 | +2.1 ± 6.6 |
| **GCN Skip-PC** | -2.0 ± 6.0 | +2.4 ± 3.3 | +2.4 ± 3.5 | **+15.6 ± 3.9** | **+9.2 ± 5.9** | **+3.6 ± 6.0** | +4.9 ± 3.0 | +15.0 ± 6.2 | +20.1 ± 9.6 |
| **MLP** | -20.4 ± 5.1 | -11.6 ± 5.8 | -6.8 ± 5.2 | -0.4 ± 3.2 | -6.8 ± 5.3 | -0.4 ± 6.5 | -9.6 ± 2.3 | -16.3 ± 4.3 | -4.2 ± 3.2 |

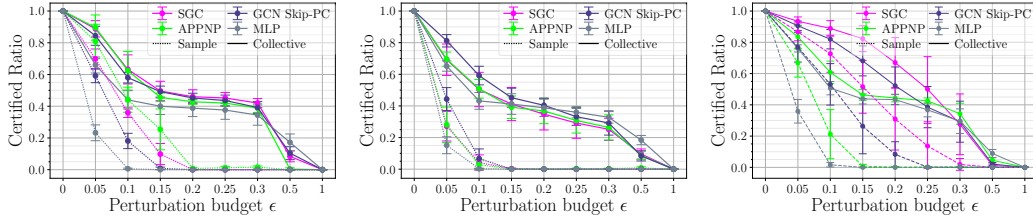

(a) Cora-MLb (Collect. & Sample.)  (b) Citeseerb (Collect. & Sample.)  (c) CSBM (Collect. & Sample.)

Figure 3: Certified ratios of selected architectures as calculated with our sample-wise and collective certificate. We refer to App. H.2.2 for collective results on all GNNs. Collective certification provides significantly higher certified ratios, and uncovers a plateauing phenomenon for intermediate $\epsilon$.

highlights: ($i$) a *clear and nearly consistent hierarchy* emerges across perturbation budgets $\epsilon$. Exemplary, for Cora-MLb (Fig. 2a) and $\epsilon = 0.05$, APPNP is most robust achieving a certified accuracy of $79.1 \pm 10.9\%$, whereas GraphSAGE achieves only $52.8 \pm 5.8\%$, and an MLP even drops to $22.7 \pm 5\%$. In addition, the rankings of the GNNs stay nearly consistent across perturbations for all datasets. ($ii$) The rankings of GNNs *differ* for each dataset. Exemplary, in contrast to Cora-MLb, the most robust model for Citeseerb is GCN Skip-PC (Fig. 2b), and for CSBM is SGC (Fig. 2c). ($iii$) Our certificate identifies *the smallest perturbation beyond which no model prediction is certifiably robust* for each dataset. From Fig. 2, the thresholds for Cora-MLb is $(0.15, 0.2]$, for Citeseerb is $(0.1, 0.15]$, and for CSBM is $(0.15, 0.2]$ except for GCN Skip-PC at $(0.2, 0.25]$ and SGC at $(0.25, 0.3]$. These findings underscore the capabilities of sample-wise certificates to provide a detailed analysis of the worst-case robustness of GNNs to label poisoning.

We now move to **collective certification** which is a more practical setting from the adversary's perspective, where the attacker can change the training dataset only once to misclassify the entire test set. Here, we demonstrate the capabilities of our Thm. 2 in certifying GNNs. In Fig. 3, we contrast the certified ratios obtained by sample-wise certification with those obtained by our collective certificate for selected architectures. They highlight a *stark contrast* between sample-wise and collective certification, with the collective certificate leading to significantly higher certified ratios, and the capability to certify even strong adversaries. Exemplary, Fig. 3a shows for the intermediate perturbation $\epsilon = 0.2$ that the sample-wise certificate cannot certify any GNN. However, the collective certificate leads to certified ratios of $> 40\%$ for all shown GNNs. This substantial difference is because the adversary is now restricted to creating only a single label perturbation to attack the entire test data, but the magnitude of the difference in certified ratios is still significant. Further, the most robust model may *not* coincide with the sample-wise case as e.g., for Cora-MLb $\epsilon = 0.1$ APPNP achieves the highest sample-wise, but from Tab. 1, SGC the highest collective robustness. This highlights the importance of collective certification to understand the worst-case robustness for the more practical scenario. In App. H.2.4, we calculate average robustness rankings for GNNs for more comprehensive $\epsilon$ ranges and show that collective robustness rankings too are *data dependent*.

Fig. 3a shows another surprising phenomenon uncovered by our collective certificate. The certified ratio seems to **plateau** for intermediate budgets $\epsilon \in [0.15, 0.3]$. Exemplary, for SGC and APPNP, the certified ratio from $\epsilon = 0.2$ to $\epsilon = 0.25$ reduces by only 0.8%, whereas the drop between

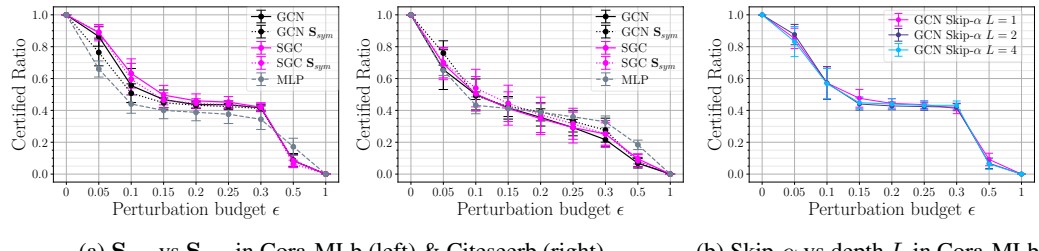

(a) $\mathbf{S}_{\text{row}}$ vs $\mathbf{S}_{\text{sym}}$ in Cora-MLb (left) & Citeseerb (right)     (b) Skip-$\alpha$ vs depth $L$ in Cora-MLb

Figure 4: Selected architectural findings based on our collective certificates. $(a)$ The effect of graph normalizations $\mathbf{S}_{\text{row}}$ and $\mathbf{S}_{\text{sym}}$ is data-dependent. $(b)$ For skip-connections, depth does not improve robustness, shown for GCN Skip-$\alpha$, see App. H.2.5 for other GNNs and datasets.

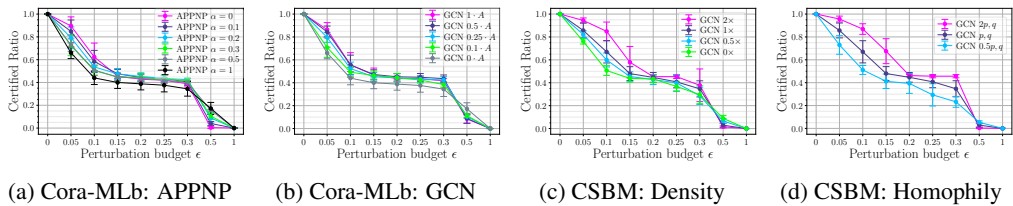

(a) Cora-MLb: APPNP     (b) Cora-MLb: GCN     (c) CSBM: Density     (d) CSBM: Homophily

Figure 5: Graph structure findings based on our collective certificates. $(a)-(b)$ The higher amount of graph information improves certifiable robustness. $(c)-(d)$ Graph density and homophily positively affect the certifiable robustness, shown for GCN using CSBM, see App. H.2.6 for more results.

$\epsilon = 0.05$ to $\epsilon = 0.01$ is 25.6% and 27.2%, respectively. The certified ratio of a GCN for $\epsilon = 0.2$ and $\epsilon = 0.25$ stays even constant (Tab. 6), as is also observed for GCN Skip-$\alpha$ (Fig. 4b). The plateau for intermediate $\epsilon$ appears for some architectures on Citeseerb, Wiki-CSb and Chameleonb, but is less pronounced, whereas Polblogsb shows near perfect plateauing (see App. H.2.3). On CSBM, SGC and GCN Skip-PC do not exhibit plateauing, while other architectures show a prominent plateau for intermediate $\epsilon$ (Fig. 3c); interestingly, a robustness plateau can be provoked by increasing the density in the graph (Figs. 5c and 5d). However, graph structure alone cannot explain the phenomenon, as Fig. 3c also shows near constancy of an MLP from $\epsilon = 0.15$ to $\epsilon = 0.2$.

Another strong observation from the sample-wise and collective certificates is the importance of graph structure in *improving* the worst-case robustness of GNNs. From the certified accuracies in Fig. 2, an MLP is always the least accurate model without any perturbation ($\epsilon = 0$), and also less robust than its GNN counterparts, as expected. Interestingly, the certified ratio plots in Fig. 3 show that MLP is consistently the least robust and the most vulnerable model for weak perturbation budgets. Thus, leveraging graph structure consistently improves sample-wise and collective robustness to label flipping, which is studied in detail in Sec. 4.2.

## 4.2 FINDINGS ON ARCHITECTURAL CHOICES AND GRAPH STRUCTURE

Leveraging our collective certificate, we investigate the influence of different architectural choices on certifiable robustness. $(I)$ *Linear activations* in GNNs are known to generalize well. Exemplary, SGC, which replaces the ReLU non-linearity in GCN to a linear activation, achieves better or similar generalization performance as a GCN, both empirically (Wu et al., 2019) and theoretically Sabanayagam et al. (2023). Complementing these results, we find that SGC is consistently better ranked than GCN across all datasets (Tab. 13), suggesting that **linear activation is as good as or better than ReLU** for certifiable robustness as well. $(II)$ Additionally, in SGC and GCN, the *graph normalization* is a design choice with $\mathbf{S}_{\text{row}}$ and $\mathbf{S}_{\text{sym}}$ being popular. While previous works (Wang et al., 2018; Sabanayagam et al., 2023) suggest that $\mathbf{S}_{\text{row}}$ leads to better generalization than $\mathbf{S}_{\text{sym}}$, our findings show that the effectiveness of these normalizations for certifiable robustness is highly dataset-dependent, as demonstrated in Fig. 4a for Cora-MLb and Citeseerb. $(III)$ *Skip-connections* in GNNs are promoted to construct GNNs with large depths as it is shown to mitigate over-smoothing (Chen et al., 2020). Our findings show that **increasing the depth in GNNs with skip-connections has either little or more pronounced negative effects on certifiable robustness**, as evidenced in Fig. 4b. For other GNNs and a more general study on depth we refer to Fig. 9.

Next, building on the importance of graph information, we conduct a deeper study into the influence of graph structure and its connectivity on certifiable robustness. $(I)$ We first explore the role of *graph input in the GNNs*: in APPNP, the $\alpha$ parameter controls the degree of graph information incorporated into the network—lower $\alpha$ implies more graph information. Similarly, in convolution-based GNNs, the graph structure matrix $\mathbf{S}$ in GCN can be computed using weighted adjacency matrix $\beta\mathbf{A}$. These experiments clearly confirm that **increasing the amount of graph information improves certifiable robustness up to intermediate attack budgets**, as demonstrated for Cora-MLb in Figs. 5a and 5b. Interestingly, for stronger budgets, the observation changes where more graph information hurts certifiable robustness, a pattern similarly observed in Gosch et al. (2024) for feature poisoning using an incomplete sample-wise certificate. $(II)$ We then analyze the effect of *graph density and homophily* by taking advantage of the random graph models. To assess graph density, we proportionally vary the density of connections within ($p$) and outside ($q$) the classes, while for homophily, we vary only $p$ keeping $q$ fixed. The results consistently show that **higher graph density and increased homophily improves certifiable robustness** with an inflection point for stronger budgets as observed in Figs. 5c and 5d. Additionally, our results generalize to changing the number of labeled nodes (App. H.2.7) and to dynamic graphs that evolve over time (App. H.4).

## 5 CONCLUSION

By leveraging the NTK that describes the training dynamics of wide neural networks, we introduce the first exact certificate for label flipping applicable to NNs. Crucially, we develop not only sample-wise but also collective certificates, and establish several significant takeaways by evaluating a broad range of GNNs on different node classification datasets:

> **Key Takeaways on Certifying GNNs Against Label Poisoning**
>
> 1. There is no silver bullet: robustness hierarchies of GNNs are **strongly data dependent**.
> 2. **Collective certificates complement sample-wise**, providing a holistic picture of the worst-case robustness of models.
> 3. **Certifiable robustness plateaus** at intermediate perturbation budgets.
> 4. **Linear activation helps**, and **depth in skip-connections hurts** certifiable robustness.
> 5. **Graph structure helps** improving robustness against label poisoning.

Among the results, the intriguing plateauing of certifiable robustness in collective evaluation has so far not been observed. While we conduct a preliminary experimental investigation, the cause of the plateauing is still unclear, and a more rigorous analysis remains an open avenue for future research.

**Generality of our certification framework.** Our certification strategy extends beyond GNNs and applies to general wide NNs through their NTKs and any kernelized SVM. Exemplary, we demonstrate the applicability to an MLP in Sec. 4 and to a linear kernel $\mathbf{X}\mathbf{X}^T$ where $\mathbf{X}$ is the feature matrix (a non-NN based model) in App. H.3. In addition, since our certificates leverage the NTK of the NN, they hold with respect to expectation over network initializations. As a result, they provide guarantees at the population level of the parameters, thus certifying NN for general parameterization. This distinguishes our framework from most certification methods, which typically focus on guarantees for a specific, fixed network parameterization.

**Scalability.** Exact certification, even for the much simpler case of test-time attacks, where the model to be certified is *fixed*, is already NP-hard (Katz et al., 2017). Thus, it is inherently difficult and a current, unsolved problem to scale exact certificates to large datasets. In fact, state-of-the-art exact certificates against test-time (evasion) attacks for image classification scale up to CIFAR-10 (Li et al., 2023), and for GNNs to graphs the size of Citeseer (Hojny et al., 2024). Similarly, we find that the scaling limits of our certificates are graphs the size of Cora-ML or Citeseer, even though the exact certification of poisoning attacks adds additional complexity with the model being certified is *not fixed* and the training dynamics must be included in the certification. As a first step toward improving scalability, we introduce a more scalable but incomplete multi-class certification strategy in App. A.2 that relaxes exactness. Here, we observe that the results are surprisingly close to and often even match the exact certificate. This demonstrates the effectiveness of our relaxation and highlights that further analysis into its tightness is an interesting topic for future study.

# 6 ETHICS STATEMENT

Our work allows for the first time exact quantification of the worst-case robustness of different (wide) GNNs to label poisoning. While a potentially malicious user could misuse these insights, we are convinced that understanding the robustness limitations of neural networks in general and GNNs, in particular, is crucial to enable a safe deployment of these models in the present and future. Thus, we believe the potential benefits of robustness research outweigh its risks. Additionally, we do not see any immediate risk stemming from our work.

# 7 REPRODUCIBILITY STATEMENT

We undertook great efforts to make our results reproducible. In particular, the experimental details are outlined in detail in Sec. 4 and App. G. All chosen hyperparameters are listed in App. G. Randomness in all experiments is controlled through the setting of seeds in involved pseudorandom number generators. The code to reproduce our results, including all experimental configuration files, can be found at `https://github.com/saper0/qpcert`.

## ACKNOWLEDGEMENTS

The authors thank Aleksei Kuvshinov, Arthur Kosmala and Pascal Esser for the helpful feedback on the manuscript. This paper has been supported by the DAAD programme Konrad Zuse Schools of Excellence in Artificial Intelligence, sponsored by the German Federal Ministry of Education and Research; by the German Research Foundation, grant GU 1409/4-1; as well as by the TUM Georg Nemetschek Institute Artificial Intelligence for the Built World.

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

## A   MULTI-CLASS LABEL CERTIFICATION

To generalize the binary classification setting in Sec. 2 to multi-class classification, we use a one-vs-all classification approach. This means, given $K$ classes, $K$ binary learning problems are created, one corresponding to each class $c \in [K]$, where the goal is to correctly distinguish instances of class $c$ from the other classes $c' \in [K]$, $c' \neq c$, which are collected into one "rest" class. Assume that $p_c$ is the prediction score of a classifier for the learning problem corresponding to class $c$. Then, the class prediction $c^*$ for a node is constructed by $c^* = \arg\max_{c \in [K]} p_c$.

In the following, we first present an exact multi-class certificate in App. A.1 and present a relaxed (incomplete) certificate in App. A.2 that significantly improves computational speed while being empirically nearly as tight as the exact certificate.

### A.1   EXACT CERTIFICATE

For the following development of an exact certificate for the multi-class case, we assume an SVM given in its dual formulation (Eq. (2)) as our model. Further, without loss of generality, assume for a learning problem corresponding to class $c$ that nodes having class $c$ will get label $1$ and nodes corresponding to the other classes $c' \in [K]$, $c' \neq c$ have label $-1$. We collect the labels for the learning problem associated to $c$ in the vector $\mathbf{y}^c$. The original multi-class labels are collected in the vector $\mathbf{y}$. Thus, one $\mathbf{y}$ defines a tuple $(\mathbf{y}^1, \dots, \mathbf{y}^K)$. Thus, any $\tilde{\mathbf{y}} \in \mathcal{A}(\mathbf{y})$ spawns a perturbed tuple $(\tilde{\mathbf{y}}^1, \dots, \tilde{\mathbf{y}}^K)$. We denote by $\hat{c}$ the originally predicted class with prediction score $p_{\hat{c}}$.

To know whether the prediction can be changed by a particular $\tilde{\mathbf{y}} \in \mathcal{A}(\mathbf{y})$, we need to know if $p_{\hat{c}} - \max_{c \in [K] \setminus \{\hat{c}\}} p_c$ can be forced to be smaller than $0$ for any $\tilde{\mathbf{y}} \in \mathcal{A}(\mathbf{y})$. By collecting the individual predictions $p_c$ in a vector $\mathbf{p} \in \mathbb{R}^K$ this problem can be formulated as the following optimization problem:

$$M_1(\mathbf{y}): \min_{\mathbf{p}, p^*, \tilde{\mathbf{y}}^1, \dots, \tilde{\mathbf{y}}^K} p_{\hat{c}} - p^* \tag{13}$$

$$s.t. \quad p^* = \max_{c \in [K] \setminus \{\hat{c}\}} p_c \tag{14}$$

$$\forall c \in [K]: \ p_c = \max_{\boldsymbol{\alpha}^c} \sum_{i=1}^m \tilde{y}_i^c \alpha_i^c Q_{ti} \quad s.t. \quad \boldsymbol{\alpha}^c \in \mathcal{S}(\tilde{\mathbf{y}}^c) \tag{15}$$

$$(\tilde{\mathbf{y}}^1, \dots, \tilde{\mathbf{y}}^K) \in \mathcal{A}\left((\mathbf{y}^1, \dots, \mathbf{y}^K)\right) \tag{16}$$

where we represented $\tilde{\mathbf{y}} \in \mathcal{A}(\mathbf{y})$ in Eq. (16) equivalently by the labels $\mathbf{y}^c$ defined for each of the $K$ learning problems. First note that $M_1(\tilde{\mathbf{y}})$ defines a complicated (trilevel) optimization problem where Eq. (15) defines $K$ bilevel problems that are independent of one another. However, observe that the objective in Eq. (15) represents the prediction of an SVM for node $t$ and thus, $\sum_{i=1}^m \tilde{y}_i^c \alpha_i^c Q_{ti}$ has the same value for any choice of $\boldsymbol{\alpha}^c \in \mathcal{S}(\tilde{\mathbf{y}}^c)$, i.e., while the optimal dual variables are not unique, the prediction value is. As a result, problem $M_1(\tilde{\mathbf{y}})$ can be written equivalently as

$$M_2(\mathbf{y}): \min_{\mathbf{p}, p^*, \tilde{\mathbf{y}}^1, \dots, \tilde{\mathbf{y}}^K} p_{\hat{c}} - p^* \tag{17}$$

$$s.t. \quad p^* = \max_{c \in [K] \setminus \{\hat{c}\}} p_c \tag{18}$$

$$\forall c \in [K]: \ p_c = \sum_{i=1}^m \tilde{y}_i^c \alpha_i^c Q_{ti} \tag{19}$$

$$\boldsymbol{\alpha}^c \in \mathcal{S}(\tilde{\mathbf{y}}^c) \tag{20}$$

$$(\tilde{\mathbf{y}}^1, \dots, \tilde{\mathbf{y}}^K) \in \mathcal{A}\left((\mathbf{y}^1, \dots, \mathbf{y}^K)\right) \tag{21}$$

Problem $M_2(\mathbf{y})$ now corresponds to a bilevel problem with $K$ inner problems $\boldsymbol{\alpha}^c \in \mathcal{S}(\tilde{\mathbf{y}}^c)$ that are independent of one another. As the inner problems are independent of one another, $M_2(\mathbf{y})$ can

actually be written as a single bilevel problem with a single inner-problem that decomposes into $K$ independent problems. This can be seen by the fact that solving the individual dual problems $\boldsymbol{\alpha}^c \in \mathcal{S}(\tilde{\mathbf{y}}^c)$ for all $c \in [K]$ is equivalent to solving the following single optimization problem:

$$\min_{(\boldsymbol{\alpha}^1,\ldots,\boldsymbol{\alpha}^K)} -\sum_{c=1}^{K}\left[\sum_{i=1}^{m}\alpha_i^c + \frac{1}{2}\sum_{i=1}^{m}\sum_{j=1}^{m}y_i^c y_j^c \alpha_i^c \alpha_j^c Q_{ij}\right] \text{ s.t. } 0 \leq \alpha_i^c \leq C \;\forall i \in [m] \wedge c \in [K] \quad (22)$$

Let's denote the set of optimal solution to Eq. (22) similarly as $\mathcal{S}\left((\mathbf{y}^1,\ldots,\mathbf{y}^K)\right)$. Then, we can rewrite $M_2(\mathbf{y})$ as follows:

$$M_3(\tilde{\mathbf{y}}): \min_{\mathbf{p},p^*,\tilde{\mathbf{y}}^1,\ldots,\tilde{\mathbf{y}}^K} p_{\hat{c}} - p^* \quad (23)$$

$$s.t. \quad p^* = \max_{c \in [K]\setminus\{\hat{c}\}} p_c \quad (24)$$

$$p_c = \sum_{i=1}^{m} \tilde{y}_i^c \alpha_i^c Q_{ti} \quad \forall c \in [K] \quad (25)$$

$$(\boldsymbol{\alpha}^1,\ldots,\boldsymbol{\alpha}^K) \in \mathcal{S}\left((\mathbf{y}^1,\ldots,\mathbf{y}^K)\right) \quad (26)$$

$$(\tilde{\mathbf{y}}^1,\ldots,\tilde{\mathbf{y}}^K) \in \mathcal{A}\left((\mathbf{y}^1,\ldots,\mathbf{y}^K)\right) \quad (27)$$

Now, we want to formulate the bilevel problem $M_3(\mathbf{y})$ as a MILP. For this, we have to address how to linearly model Eqs. (24), (26) and (27) in the following.

**Inner Problem.** To linearly model Eq. (26), recognize that it still is a convex optimization problem fulfilling Slater's condition by the same argumentation as provided in App. B. Thus, we can replace $(\boldsymbol{\alpha}^1,\ldots,\boldsymbol{\alpha}^K) \in \mathcal{S}\left((\mathbf{y}^1,\ldots,\mathbf{y}^K)\right)$ with its KKT conditions and won't change the optimal solutions to the optimization problem. The KKT conditions of $(\boldsymbol{\alpha}^1,\ldots,\boldsymbol{\alpha}^K) \in \mathcal{S}\left((\mathbf{y}^1,\ldots,\mathbf{y}^K)\right)$ turn out to be the KKT conditions of the individually involved subproblems $\boldsymbol{\alpha}^c \in \mathcal{S}(\tilde{\mathbf{y}}^c)$ for all $c \in [K]$ due to their independence from one another. Then, the exact same linear modeling strategy for the stationarity and complementary slackness constraints can be leveraged, as introduced in Sec. 3.1. Note that we now get class-dependent dual variables $v_i^c$ and $u_i^c$ as well as class-dependent binary variables $s_i^c$ and $t_i^c$ for each labeled node $i$ and class $c$. Thus, we have successfully linearized Eq. (26) and for all constraints written out, we refer to Thm. 3.

**Max.** To model the $\max$ in Eq. (24) note that $p^L = -C\sum_{i=1}^{m}|Q_{ti}|$ and $p^U = C\sum_{i=1}^{m}|Q_{ti}|$ define a lower and upper bound to $p_c$, respectively, valid for all $c \in [K]$. Now, the maximum constraint Eq. (24) can be linearly modeled introducing a binary variable $\mathbf{b} \in \{0,1\}^{K-1}$ with

$$\sum_{c \in [K]\setminus\{\hat{c}\}} b_c = 1 \wedge \forall c \in [K]\setminus\{\hat{c}\}: p^* \geq p_c, \; p^* \leq p_c + (1-b_k)(p^U - p^L), \; b_c \in \{0,1\} \quad (28)$$

**Adversary.** Lastly, we have to linearly model the adversary in Eq. (27). For this, we represent the original labels in a new vector $\mathbf{y}'^c \in \{0,1\}^m$ with $y_i'^c = 1$ if $y_i^c = 1$ and 0 otherwise for all classes $c \in [K]$ and labeled nodes $i \in [m]$. Similarly, we introduce binary variables $\tilde{\mathbf{y}}'^c \in \{0,1\}^m$ with $\tilde{y}_i'^c = 1$ if $\tilde{y}_i^c = 1$ and 0 otherwise. Our goal is to use $\tilde{\mathbf{y}}'^c$ to define $\tilde{\mathbf{y}}^c$ and at the same time model the adversaries' strength. Setting $\tilde{\mathbf{y}}^c$ can be achieved by the linear constraint $\tilde{y}_i^c = 2\tilde{y}_i'^c - 1$ for all $i \in [m]$ and $c \in [K]$. As a node $i$ can only be in one class, $\tilde{y}_i'^c$ can only be 1 for one $c$ and has to be 0 for all other classes. This can be ensured by the constraint $\sum_{c=1}^{K} \tilde{y}_i'^c = 1$. Lastly, the adversaries' strength can be modeled using

$$\sum_{i=1}^{m}\left(1 - \sum_{c=1}^{K} y_i'^c \tilde{y}_i'^c\right) \leq \lfloor \epsilon m \rfloor \quad (29)$$

Thus, we have now successfully shown how to linearly model Eqs. (24), (26) and (27) and can state the following theorem:

**Theorem 3 (Multiclass MILP)** *Given the adversary $\mathcal{A}$, positive constants $M_{u_i}^c$ and $M_{v_i}^c$ set as in App. C for all $i \in [m] \wedge c \in [K]$, and $p^L, p^U \in \mathbb{R}$ with $p^L = -C \sum_{i=1}^m |Q_{ti}|$ and $p^U = C \sum_{i=1}^m |Q_{ti}|$, the prediction for node $t$ is certifiably robust if the optimal solution to the MILP $M(\mathbf{y})$, given below, is greater than zero and non-robust otherwise.*

$$M(\mathbf{y}): \min_{\substack{p^*, \mathbf{p}, \boldsymbol{s}, \boldsymbol{t}, \mathbf{R}, \boldsymbol{z}, \boldsymbol{b}, \boldsymbol{\alpha}, \\ \tilde{\mathbf{y}}^1, \dots, \tilde{\mathbf{y}}^K, \tilde{\mathbf{y}}'^1, \dots, \tilde{\mathbf{y}}'^K}} p_{\hat{c}} - p^* \quad s.t.$$

$$\sum_{i=1}^m \left(1 - \sum_{c=1}^K y_i'^c \tilde{y}_i'^c\right) \le \lfloor \epsilon m \rfloor, \ \sum_{c=1}^K \tilde{y}_i'^c = 1, \ \sum_{c \in [K] \setminus \{\hat{c}\}} b_c = 1$$

$$\forall c \in [K] \setminus \{\hat{c}\}: p^* \ge p_c, \ p^* \le p_c + (1 - b_k)(p^U - p^L), \ b_c \in \{0, 1\}$$

$$\forall c \in [K]: p_c = \sum_{i=1}^m z_i^c Q_{ti}$$

$$\forall c \in [K] \wedge i, j \in [m]: \sum_{j=1}^m R_{ij}^c Q_{ij} - 1 - u_i^c + v_i^c = 0$$

$$- C(1 + \tilde{y}_i^c) \le R_{ij}^c + z_j^c \le C(1 + \tilde{y}_i^c)$$

$$- C(1 - \tilde{y}_i^c) \le R_{ij}^c - z_j^c \le C(1 - \tilde{y}_i^c),$$

$$\forall c \in [K] \wedge i \in [m]: -\alpha_i^c \le z_i^c \le \alpha_i^c, \quad \alpha_i^c - C(1 - \tilde{y}_i^c) \le z_i^c \le C(1 + \tilde{y}_i^c) - \alpha_i^c$$

$$u_i \le M_{u_i}^c s_i^c, \ \alpha_i^c \le C(1 - s_i^c), \ v_i \le M_{v_i}^c t_i^c, \ \alpha_i^c \ge C t_i^c,$$

$$s_i^c \in \{0, 1\}, \ t_i^c \in \{0, 1\}, \quad u_i^c \ge 0, \quad v_i^c \ge 0$$

$$\tilde{y}_i^c = 2\tilde{y}_i'^c - 1, \ \tilde{y}_i'^c \in \{0, 1\}$$

**Computation Complexity.** The MILP has $3Km + K - 1$ binary variables.

## A.2 INEXACT CERTIFICATE

Thm. 1 can be extended to an incomplete multi-class certificate by a strategy similarly proposed in Gosch et al. (2024). Assume that $c^*$ is the original prediction of our model without poisoning. Now, we solve an optimization problem very similar to $P(\mathbf{y})$ from Thm. 1 for each learning problem defined by $c \in [K]$, but change the objective $\min \text{sign}(\hat{p}_t) \sum_{i=1}^m z_i Q_{ti}$ either to $\min \sum_{i=1}^m z_i Q_{ti}$ if $c = c^*$ or $\max \sum_{i=1}^m z_i Q_{ti}$ if $c \neq c^*$. Then, the original prediction is certifiably robust, if the solution to the minimization problem is still larger than the maximum solution to any maximization problem.

**Tightness.** On first sight, one might expect that such a strategy trades of tightness significantly with scalability. However, we surprisingly find on Citeseer that the inexact certificate provides mostly the same certified accuracy, or a certified accuracy only very marginally below the exact certificate as shown in Tab. 2. We hypothesise that this is due to the fact that the most effective label perturbation may be one that flips the training labels of the predicted class to the runner-up (second highest logit score) class until the budget is exhausted. While the relaxation allows for independent changes to each of the $K$ classifiers until the budget is used up, following the above argumentation, being able to make these independent changes to the other classes (not the highest predicted or second-highest) may not make the adversary significantly stronger.

**Scalability.** The exact multi-class certificate introduces $K$ times as many binary variables as the binary certificate. The inexact certificate manages to reduce the multi-class certification problem to a binary one and thereby significantly improve scalability. Exemplary, for Cora-ML, the multi-class certificate is not computable in reasonable runtimes due to the larger $C$ values of the different GNNs. However, running the incomplete certificate makes the problem running in a reasonable time with an average certification runtime of around 20 minutes per node (using two CPUs) with some rare cases taking up to 6h. The results for Cora-ML can be found in App. I.

Table 2: Certified accuracy in [%] for Citeseer of the exact vs incomplete multiclass certificate. $\epsilon \in \{0, 0.01, 0.02, 0.03, 0.05\}$ refers to the attack budget (fraction of flipped labels in the training set.)

|  | $\epsilon$ | 0 (Clean Acc.) | 0.01 | 0.02 | 0.03 | 0.05 |
|---|---|---|---|---|---|---|
| GCN | Exact | $66 \pm 3.3$ | $10.7 \pm 2.5$ | $0 \pm 0$ | $0 \pm 0$ | $0 \pm 0$ |
|  | Incomplete | $66 \pm 3.3$ | $10.7 \pm 2.5$ | $0 \pm 0$ | $0 \pm 0$ | $0 \pm 0$ |
| SGC | Exact | $65.3 \pm 4.1$ | $35.3 \pm 5$ | $9.3 \pm 5.2$ | $0.7 \pm 0.9$ | $0 \pm 0$ |
|  | Incomplete | $65.3 \pm 4.1$ | $34.7 \pm 5.7$ | $8.7 \pm 4.7$ | $0.7 \pm 0.9$ | $0 \pm 0$ |
| GCN Skip-$\alpha$ | Exact | $63.3 \pm 5.2$ | $20 \pm 5.9$ | $2 \pm 1.6$ | $0 \pm 0$ | $0 \pm 0$ |
|  | Incomplete | $63.3 \pm 5.2$ | $20 \pm 5.9$ | $2 \pm 1.6$ | $0 \pm 0$ | $0 \pm 0$ |
| GCN Skip-PC | Exact | $66.7 \pm 1.9$ | $20.3 \pm 4.7$ | $0.7 \pm 0.9$ | $0 \pm 0$ | $0 \pm 0$ |
|  | Incomplete | $66.7 \pm 1.9$ | $20.3 \pm 4.7$ | $0.7 \pm 0.9$ | $0 \pm 0$ | $0 \pm 0$ |

## B  SLATER CONDITION

It is generally known that the SVM dual problem is a convex quadratic program. We now show that the SVM dual problem $P_1(\tilde{\mathbf{y}})$ in $\alpha \in \mathcal{S}(\tilde{\mathbf{y}})$ fulfills (strong) *Slater's condition*, which is a constraint qualification for convex optimization problems, for any choice of $\tilde{\mathbf{y}} \in \mathcal{A}(\mathbf{y})$. This allows to reformulate the bilevel problem in Sec. 3.1 to be reformulated into a single-level problem with the same globally optimal solutions (Dempe & Dutta, 2012). Our argumentation is similar to (Gosch et al., 2024) and adapted to the label-flipping case.

First, we define Slater's condition for the SVM problem:

**Def. 1 (Slater's condition)** *A convex optimization problem $P_1(\tilde{\mathbf{y}})$ fulfills strong Slater's Constraint Qualification, there exists a point $\boldsymbol{\alpha}$ in the feasible set of $P_1(\tilde{\mathbf{y}})$ such that no constraint in $P_1(\tilde{\mathbf{y}})$ is active, i.e. $0 < \alpha_i < C$ for all $i \in [m]$.*

**Proposition 1** *$P_1(\tilde{\mathbf{y}})$ fulfills Slater's condition for any choice $\tilde{\mathbf{y}} \in \mathcal{A}(\mathbf{y})$.*

*Proof.* It is easy to see that for a given fixed $\tilde{\mathbf{y}}$, Slater's condition holds: choose $\alpha_i = C/2$ for all $i \in [m]$, this is a feasible (but not optimal) solution with no active constraints. That Slater's condition for $P_1(\tilde{\mathbf{y}})$ holds for any $\tilde{\mathbf{y}} \in \mathcal{A}(\mathbf{y})$ can again be seen by noting, that the feasible solution defined by setting $\alpha_i = C/2$ for all $i \in [m]$ is independent of a given $\tilde{\mathbf{y}}$ and and stays a feasible solution without active constraints for any choice of $\tilde{\mathbf{y}}$. $\square$

## C  BIG-M

**Proposition 1** *Replacing the complementary slackness constraints Eq. (6) in $P_3(\mathbf{y})$ with the big-M constraints given in Eq. (10) does not cut away solution values of $P_3(\mathbf{y})$, if for all $i \in [m]$, the big-M values are set following Eqs. (30) and (31).*

$$M_{u_i} = \sum_{j=1}^{m} C|Q_{ij}| - 1 \tag{30}$$

$$M_{v_i} = \sum_{j=1}^{m} C|Q_{ij}| + 1 \tag{31}$$

*Furthermore, Eqs. (30) and (31) define the tightest possible big-M values.*

*Proof.* The proof strategy follows Gosch et al. (2024) and is adapted to the label flipping case. First, we lower and upper bound the term $\sum_{j=1}^{m} R_{ij} Q_{ij}$ for any $i \in [m]$ in the stationarity constraints. As $R_{ij} = \tilde{y}_i z_j$ and $-C \leq z_j \leq C$ and $\tilde{y}_i \in \{-1, 1\}$, it follows that $LB = -\sum_{j=1}^{m} C|Q_{ij}| \leq \sum_{j=1}^{m} R_{ij} Q_{ij} \leq \sum_{j=1}^{m} C|Q_{ij}| = UB$. It is easy to see, that the bounds are tight.

Now, the dual variable $u_i$ and $v_i$ are coupled with the other variables in the overall MILP only through the stationarity constraints $\sum_{j=1}^{m} R_{ij} Q_{ij} - 1 - u_i + v_i$ for all $i \in [m]$ and do not feature

in the objective of $P_3(rvy)$. Thus, we only have to ensure that any upper bound on $u_i$ or $v_i$, cannot affect any optimal choice for the other optimization variables. This is achieved if no feasible choice of $R_{ij}$ is cut from the solution space, which in turn is guaranteed, if any bound on $u_i$ or $v_i$, still allow the term $\sum_{j=1}^m R_{ij}Q_{ij}$ in the stationarity constraint, to take any value between $LB$ and $UB$. Using these bounds, we get

$$UB - u_i + v_i \geq 1 \tag{32}$$

$$LB - u_i + v_i \leq 1 \tag{33}$$

For the first inequality, assume $UB > 1$, then by setting $v_i = 0$ and $u_i \leq UB - 1$ fullfils all constraints and does not cut away any solution value. Similarly, if $UB < 1$, set $u_i = 0$ and $v_i \leq 1 - UB$. For the second inequality, for $LB > 1$ set $v_i = 0$ and $u_i \leq LB - 1$ and for $LB < 1$ set $u_i = 0$ and $v_i \leq 1 - LB$. By only enforcing the so mentioned least constraining bounds for $u_i$ and $v_i$, we exactly arrive at Eqs. (30) and (31) where tightness follows from the tighness of the bounds. □

## D   COLLECTIVE CERTIFICATE

We present the full version of the collective certificate Thm. 2 here.

**Theorem 4 (MILP Formulation)** *Given the adversary $\mathcal{A}$, positive constants $M_{u_i}$ and $M_{v_i}$ set as in App. C for all $i \in [m]$, and $\mathbf{l}$ and $\mathbf{h} \in \mathbb{R}^{|\mathcal{T}|}$ with $l_t = -C\sum_{i=1}^m Q_{ti}$ and $h_t = C\sum_{i=1}^m Q_{ti}$, the maximum number of test nodes that are certifiably non-robust is given by the MILP $C(\mathbf{y})$.*

$$
\begin{aligned}
C(\mathbf{y}): \quad & \max_{\substack{\boldsymbol{\alpha}, \tilde{\mathbf{y}}, \mathbf{y}', \mathbf{z}, \\ \mathbf{u}, \mathbf{v}, \mathbf{s}, \mathbf{t}, \mathbf{R}, \mathbf{c}}} \sum_{t \in \mathcal{T}} c_t \quad s.t. \quad p_t = \sum_{i=1}^m z_i Q_{ti}, \ \sum_{i=1}^m 1 - y_i \tilde{y}_i \leq 2\lfloor \epsilon m \rfloor, \ \forall i \in [m]: \tilde{y}_i = 2y_i' - 1, \\
& \forall t \in \mathcal{T}: c_t = \{0, 1\}, \quad \forall \hat{p}_t > 0 : p_t \leq h_t(1 - c_t), \ p_t > l_t c_t, \\
& \hspace{4cm} \forall \hat{p}_t < 0 : p_t \geq l_t(1 - c_t), \ p_t < h_t c_t, \\
& \forall i, j \in [m]: \sum_{j=1}^m R_{ij}Q_{ij} - 1 - u_i + v_i = 0, \quad u_i \geq 0, \quad v_i \geq 0, \quad y_i' \in \{0, 1\}, \\
& \hspace{1cm} - C(1 + \tilde{y}_i) \leq R_{ij} + z_j \leq C(1 + \tilde{y}_i), \quad -C(1 - \tilde{y}_i) \leq R_{ij} - z_j \leq C(1 - \tilde{y}_i), \\
& \hspace{1cm} - \alpha_i \leq z_i \leq \alpha_i, \quad \alpha_i - C(1 - \tilde{y}_i) \leq z_i \leq C(1 + \tilde{y}_i) - \alpha_i, \\
& \hspace{1cm} u_i \leq M_{u_i} s_i, \ \alpha_i \leq C(1 - s_i), \ v_i \leq M_{v_i} t_i, \ \alpha_i \geq C t_i, \ s_i \in \{0, 1\}, \ t_i \in \{0, 1\}.
\end{aligned}
$$

## E   FINITE-WIDTH MODEL-SPECIFIC GUARANTEES

We derive model-specific guarantees for finite-width setting that includes the depth, width, and activation functions used. To obtain this, we follow the derivation in Liu et al. (2020); Chen et al. (2021) and consider normalized input node features, bounded spectral norm of the graph convolution, Lipschitz and smooth activation function. Concretely, we consider a graph neural network with depth $L$, width $w$ and activation function with Lipschitz constant $\rho$, and trained using regularized Hinge loss with $C$ as the regularization constant. Let the network parameters $W$ during training move within a fixed radius $R > 0$ to initialization $W_{init}$, i.e. $\{W \mid ||W - W_{init}|| \leq R\}$. Then, the output difference between an infinite-width network and a finite-width network is determined by the deviation of the finite-width NTK at time $t$ from the NTK at initialization, similar to standard neural networks (Chen et al., 2021, Section F.1). Now, this NTK deviation is determined by the Hessian spectral norm of the network as shown in Liu et al. (2020). Thus, we bound the Hessian spectral norm by bounding the parameters, each layer outputs, their gradients and second-order gradients. Since we consider the node features $\mathbf{X}$ are normalized and the spectral norm of graph convolution $\mathbf{S}$

is bounded[2], we get the Hessian spectral norm to be bounded as $\mathcal{O}(\frac{R^{3L+1}\ln w}{w})$. Consequently, using this, we get the bound for the output difference between an infinite-width network and a finite-width network as $\mathcal{O}(\frac{R^{3L+1}\rho \ln w}{Cw})$ with probability $p = 1 - L\exp(-\Omega(w))$. This is the same as the bounds in Liu et al. (2020). Note that this theoretical bound is not directly computable unless constants in the derivation are preserved and applied to specific inputs. Unfortunately, the literature on the NTK so far is mainly concerned with providing convergence statements in big-O notation and not with calculating the individually involved constants. Thus, it is an interesting open question to derive the explicit constants involved in the bounds.

## F    COMPARISON TO QPCERT GOSCH ET AL. (2024)

While Gosch et al. (2024) also reformulates the bilevel problem associated to data feature poisoning using the SVM equivalence, similar to our approach on a high level, the technical challenges and resulting contributions are fundamentally different, as outlined below: $(i)$ **Difference in adversary:** Gosch et al. (2024) addresses the feature poisoning setting, whereas we focus on a different problem of label poisoning. $(ii)$ **Difference in the final outcome:** While Gosch et al. (2024) derives an incomplete sample-wise certificate, we derive exact certificates for both sample-wise and collective cases. Note that collective certificates are as important as sample-wise certificates as substantially established in Sec. 4.$(iii)$ **Technical differences:** In Gosch et al. (2024), the single-level reformulation is a bilinear optimization (product of two continuous variables). As a product between two continuous variables can't be modeled exactly in a linear way, Gosch et al. (2024) relax the original optimization problem resulting in the incompleteness of their certificate. In contrast, our single-level reformulation is a nonlinear optimization, involving products of a continuous variable with two binary variables (Eq. (9)), along with bilinear terms (Eqs. (4) and (6)). These distinctions make the techniques in Gosch et al. (2024) not applicable to our problem. However, the techniques we introduce in Sec. 3.1 allow us to model these new non-linearities linearly in an exact fashion, resulting in an exact certificate.

## G    EXPERIMENTAL DETAILS

**Datasets.** We consider multi-class Cora-ML and Citeseer. Using these, we create binary datasets, Cora-MLb and Citeseerb. In addition we generate synthetic datasets using CSBM and CBA random graph models. In Tab. 3, we provide the statistics for the datasets.

| Dataset | # Nodes | # Edges | # Classes |
|---|---|---|---|
| Cora-ML | 2,810 | 7,981 | 7 |
| Cora-MLb | 1,245 | 2,500 | 2 |
| Citeseer | 2,110 | 3,668 | 6 |
| Citeseerb | 1,239 | 1,849 | 2 |
| Wiki-CSb | 4,660 | 72,806 | 2 |
| Polblogs | 1,222 | 16,714 | 2 |
| Corab | 1,200 | 1,972 | 2 |
| Chameleonb | 294 | 1,182 | 2 |
| CSBM | 200 | 367±16 | 2 |
| CBA | 200 | 389±3 | 2 |

Table 3: Dataset statistics

### G.1    GENERATING GRAPHS FROM RANDOM GRAPH MODELS

**CSBM.** A CSBM graph $\mathcal{G}$ with $n$ nodes is iteratively sampled as (a) Sample label $y_i \sim Bernoulli(1/2) \; \forall i \in [n]$; (b) Sample feature vectors $\mathbf{X}_i | y_i \sim \mathcal{N}(y_i \boldsymbol{\mu}, \sigma^2 \mathbf{I}_d)$; (c) Sample adjacency $A_{ij} \sim Bernoulli(p)$ if $y_i = y_j$, $A_{ij} \sim Bernoulli(q)$ otherwise, and $A_{ji} = A_{ij}$. Following prior work Gosch et al. (2023), we set $p, q$ through the maximum likelihood fit to Cora (Sen et al.,

---

[2]The spectral norm of $\mathbf{S}$ is $\leq 1$ for all practically used convolutions.

2008) ($p = 3.17\%$, $q = 0.74\%$), and $\boldsymbol{\mu}$ element-wise to $K\sigma/2\sqrt{d}$ with $d = \lfloor n/\ln^2(n) \rfloor$, $\sigma = 1$, and $K = 1.5$, resulting in an interesting classification scheme where both graph structure and features are necessary for good generalization.

**CBA.** Similar to CSBM, we sample nodes in a graph $\mathcal{G}$ using CBA following Gosch et al. (2023). The iterative process for each node $i \in [n]$ follows: (a) Sample label $y_i \sim Bernoulli(1/2)$; (b) Sample feature vectors $\mathbf{X}_i|y_i \sim \mathcal{N}(y_i\boldsymbol{\mu}, \sigma^2\mathbf{I}_d)$; (c) Choose $m$ neighbors based on a multinomial distribution, where the fixed parameter $m$ is the degree of each added node. The probability of choosing neighbour $j$ is $p_j = \frac{(1+\deg_j)w_{ij}}{\sum_{m=1}^{i-1}(1+\deg_m)w_{im}}$ where $\deg_j$ is the degree of node $j$ and $w_{ij}$ is the fixed affinity between nodes $i$ and $j$ based on their class labels. When a neighbor node $j$ gets sampled more than once, we set $A_{ij} = 1$.

## G.2 HYPERPARAMETERS

We set the hyperparameters based on 4-fold cross-validation, and regarding the regularization parameter $C$, we choose the smallest one within the standard deviation of the best validation accuracy for simulated datasets and the best one based on the validation accuracy for all real datasets.

For CSBM, we choose $\mathbf{S}$ to $\mathbf{S}_{\text{row}}$ for GCN, SGC, GCN Skip-$\alpha$ and GCN Skip-PC, $\mathbf{S}_{\text{sym}}$ for APPNP with its $\alpha = 0.1$. GIN and GraphSAGE are with fixed $\mathbf{S}$. In the case of $L = 1$, the regularization parameter $C$ is 0.001 for all GNNs except APPNP where $C = 0.5$. For $L = 2$, $C = 0.001$ for all, except GCN with $C = 0.25$ and GCN Skip-$\alpha$ with $C = 0.25$. For $L = 4$, again $C = 0.001$ for all, except GCN with $C = 0.25$ and GCN Skip-$\alpha$ with $C = 0.5$.

For CBA, the best $\mathbf{S}$ is $\mathbf{S}_{\text{sym}}$ for GCN, SGC, GCN Skip-$\alpha$, GCN Skip-PC, and APPNP with its $\alpha = 0.3$. GIN and GraphSAGE are with fixed $\mathbf{S}$. In the case of $L = 1$, the regularization parameter $C$ is 0.001 for all GNNs. For $L = 2$, $C = 0.001$ for all, except GCN with $C = 0.25$ and GCN Skip-$\alpha$ with $C = 0.25$. For $L = 4$, again $C = 0.001$ for all, except GCN with $C = 0.5$ and GCN Skip-$\alpha$ with $C = 0.25$.

We outline the hyperparameters for real world datasets. All hyperparameter choices for all architecture and experiments can be found in the experiment files in the linked code.

| $C$-values | Cora-MLb | Citeseerb | Cora-ML | Citeseer |
|---|---|---|---|---|
| GCN (Row Norm.) | 0.075 | 0.75 | 0.004 | 0.0001 |
| GCN (Sym. Norm.) | 0.075 | 0.1 | - | - |
| SGC (Row Norm.) | 0.075 | 2.5 | 0.004 | 0.0001 |
| SGC (Sym Norm.) | 0.05 | 1 | - | - |
| APPNP (Sym. Norm.) | 0.5, $\alpha = 0$ | 0.5, $\alpha = 0.2$ | - | - |
| MLP | 0.025 | 0.025 | - | - |
| GCN Skip-$\alpha$ (Row Norm.) | 0.1, $\alpha = 0.1$ | 0.25, $\alpha = 0.3$ | 0.004, $\alpha = 0.2$ | 0.0001, $\alpha = 0.5$ |
| GCN SkipPC (Row Norm.) | 0.075 | 0.075 | 0.003 | 0.0001 |
| GIN | 0.025 | 0.005 | - | - |
| GraphSAGE | 0.0075 | 0.025 | - | - |

Table 4: Best Hyperparameters Real World.

| $C$-values | Wiki-CSb | Polblogs | Corab | Chameleonb |
|---|---|---|---|---|
| GCN (Row Norm.) | 1 | 10 | 0.25 | 10 |
| SGC (Row Norm.) | 0.5 | 10 | 0.1 | 0.5 |
| APPNP (Sym. Norm.) | 5, $\alpha = 0$ | - | 0.25, $\alpha = 0.2$ | 0.75, $\alpha = 0.3$ |
| MLP | 0.75 | 0.001 | 0.25 | 0.1 |
| GCN Skip-$\alpha$ (Row Norm.) | 1, $\alpha = 0.1$ | 10, $\alpha = 0.1$ | 0.5, $\alpha = 0.1$ | - |
| GCN SkipPC (Row Norm.) | 1 | 2.5 | 0.25 | - |
| GIN | 0.175 | 0.075 | 0.025 | 0.01 |
| GraphSAGE | - | 0.75 | 0.01 | 0.75 |

Table 5: Best Hyperparameters Real World.

For Cora-MLb, further, the following architectures were used with row normalization:

- GCN L=2: C=0.05

- GCN L=4: C=0.1

- GCN Skip-PC L=2: C=0.05

- GCN Skip-PC L=4: C=0.01

- GCN Skip-$\alpha$ L=2: C=0.075, $\alpha = 0.1$

- GCN Skip-$\alpha$ L=4: C=0.1, $\alpha = 0.2$

- GCN $0.25A$: C=0.05

- GCN $0.5A$: C=0.075

- GCN $0.75A$: C=0.075

We choose the best C given by 4-fold CV, except for Cora-ML, where we choose the smallest C in the standard deviation of the best validation parameters in CV.

## G.3 HARDWARE

We used Gurobi to solve the MILP problems and all our experiments are run on CPU on an internal cluster. The memory requirement to compute sample-wise and collective certificates depends on the length MILP solving process. The sample-wise certificate for Cora-MLb and Citeseerb requires less than 2 GB of RAM and has a runtime of a few seconds to minutes. For the multi-class case, the exact certificate took up to 3 GB RAM and had a runtime between 1 minute to 30 minutes. The collective certificate for Cora-MLb required between 1 to 25 GB of RAM with an average requirement of 2.8 GB. The solution time took between a few seconds, and for some rare instances up to 3 days, the average runtime was $4, 2h$. The runtime and memory requirements for collective certification on Citeseerb were similar to Cora-MLb.

# H ADDITIONAL RESULTS

## H.1 SAMPLE-WISE CERTIFICATE FOR CBA AND POLBLOGS

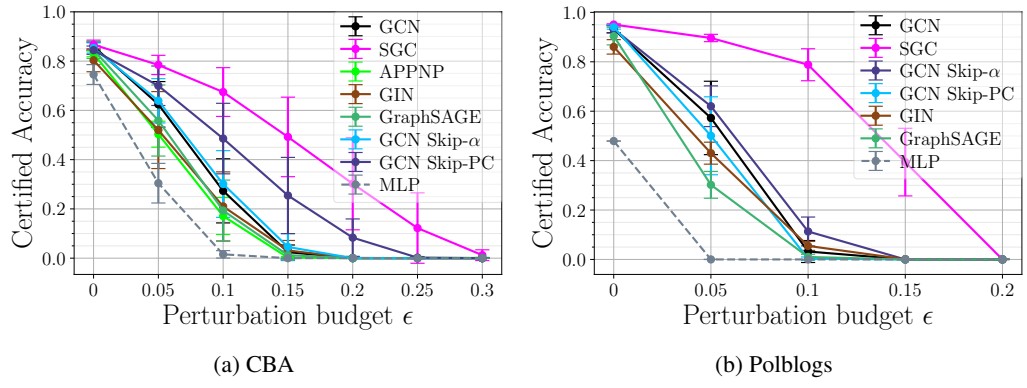

(a) CBA

(b) Polblogs

Figure 6: Certified accuracy computed with our sample-wise certificates for CBA and Polblogs datasets.

Fig. 6 shows the certified accuracy computed with our sample-wise certificates for all considered GNNs. See Fig. 2 for other datasets.

## H.2 Collective Certificate

### H.2.1 Full Certified Ratio Tables

Certified ratios for all architectures and all $\epsilon$ for Cora-MLb (Tab. 6), Citeseerb (Tab. 7), WikiCSb (Tab. 8), Polblogs (Tab. 9), Corab (Tab. 10), Chamelonb (Tab. 11), and CSBM (Tab. 12). We note that we do not report $\epsilon = 1$ as the mean certified ratio is 0 for all architectures.

Table 6: Certified ratios in [%] calculated with our exact collective certificate on **Cora-MLb** for $\epsilon \in \{0.05, 0.1, 0.15, 0.2, 0.25, 0.3, 0.5\}$. As a baseline for comparison, the certified ratio of a GCN is reported. Then, for the other models, we report their *absolute change* in certified ratio compared to a GCN, i.e., their certified ratio minus the mean certified ratio of a GCN. The most robust model for a choice of $\epsilon$ is highlighted in bold.

| $\epsilon$ | 0.05 | 0.10 | 0.15 | 0.20 | 0.25 | 0.30 | 0.50 |
|---|---|---|---|---|---|---|---|
| **GCN** | $86.4 \pm 6.1$ | $55.6 \pm 10.7$ | $46.8 \pm 6.0$ | $43.6 \pm 2.3$ | $43.6 \pm 2.3$ | $41.6 \pm 2.0$ | $8.4 \pm 3.9$ |
| **GCN** | $86.4 \pm 6.1$ | $55.6 \pm 10.7$ | $46.8 \pm 6.0$ | $43.6 \pm 2.3$ | $43.6 \pm 2.3$ | $41.6 \pm 2.0$ | $8.4 \pm 3.9$ |
| **SGC** | $+2.4 \pm 5.2$ | $\mathbf{+7.6 \pm 9.2}$ | $\mathbf{+2.8 \pm 6.1}$ | $\mathbf{+2.4 \pm 4.4}$ | $\mathbf{+1.6 \pm 3.5}$ | $\mathbf{+0.4 \pm 2.8}$ | $+0.0 \pm 3.4$ |
| **APPNP** | $\mathbf{+3.2 \pm 7.9}$ | $+6.8 \pm 12.2$ | $-1.2 \pm 2.7$ | $-0.8 \pm 1.6$ | $-1.6 \pm 1.8$ | $-2.8 \pm 2.7$ | $-8.0 \pm 0.8$ |
| **GIN** | $-4.8 \pm 4.1$ | $+6.0 \pm 4.5$ | $-2.0 \pm 5.5$ | $-5.6 \pm 5.2$ | $-6.8 \pm 5.7$ | $-8.0 \pm 6.4$ | $+4.8 \pm 3.0$ |
| **GraphSAGE** | $-6.0 \pm 6.5$ | $+1.2 \pm 5.2$ | $+1.6 \pm 6.1$ | $+1.6 \pm 4.1$ | $-1.2 \pm 3.9$ | $-4.0 \pm 3.9$ | $+3.6 \pm 2.8$ |
| **GCN Skip-$\alpha$** | $-1.2 \pm 6.4$ | $+1.6 \pm 10.2$ | $+0.8 \pm 5.6$ | $+0.8 \pm 2.9$ | $+0.0 \pm 2.3$ | $-0.4 \pm 3.2$ | $+0.8 \pm 3.9$ |
| **GCN Skip-PC** | $-2.0 \pm 6.0$ | $+2.4 \pm 3.3$ | $+2.4 \pm 3.5$ | $+1.6 \pm 3.0$ | $-0.0 \pm 3.4$ | $-2.4 \pm 3.7$ | $+2.0 \pm 4.1$ |
| **MLP** | $-20.4 \pm 5.1$ | $-11.6 \pm 5.8$ | $-6.8 \pm 5.2$ | $-4.8 \pm 5.3$ | $-6.0 \pm 5.9$ | $-7.2 \pm 6.4$ | $\mathbf{+8.8 \pm 5.3}$ |

Table 7: Certified ratios in [%] calculated with our exact collective certificate on **Citeseerb** for $\epsilon \in \{0.05, 0.1, 0.15, 0.2, 0.25, 0.3, 0.5\}$. As a baseline for comparison, the certified ratio of a GCN is reported. Then, for the other models, we report their *absolute change* in certified ratio compared to a GCN, i.e., their certified ratio minus the mean certified ratio of a GCN. The most robust model for a choice of $\epsilon$ is highlighted in bold.

| $\epsilon$ | 0.05 | 0.10 | 0.15 | 0.20 | 0.25 | 0.30 | 0.50 |
|---|---|---|---|---|---|---|---|
| **GCN** | $65.6 \pm 12.5$ | $50.0 \pm 9.7$ | $41.6 \pm 7.1$ | $35.6 \pm 5.4$ | $29.2 \pm 5.2$ | $21.6 \pm 4.1$ | $6.8 \pm 3.0$ |
| **SGC** | $+3.6 \pm 9.9$ | $+0.4 \pm 10.8$ | $-0.4 \pm 10.5$ | $-0.8 \pm 10.0$ | $-0.0 \pm 9.7$ | $3.6 \pm 7.0$ | $+2.8 \pm 3.2$ |
| **APPNP** | $+4.0 \pm 4.5$ | $+0.8 \pm 5.9$ | $-2.4 \pm 7.1$ | $+1.2 \pm 8.1$ | $+1.6 \pm 8.0$ | $+4.8 \pm 7.9$ | $+2.0 \pm 2.7$ |
| **GIN** | $+6.8 \pm 6.6$ | $+0.8 \pm 8.6$ | $+1.2 \pm 6.1$ | $\mathbf{+6.0 \pm 6.4}$ | $\mathbf{+9.6 \pm 6.8}$ | $\mathbf{+11.6 \pm 4.5}$ | $+4.0 \pm 3.2$ |
| **GraphSAGE** | $+9.6 \pm 5.6$ | $+5.2 \pm 5.3$ | $+2.4 \pm 5.2$ | $+4.4 \pm 4.9$ | $+7.6 \pm 4.7$ | $+8.8 \pm 6.6$ | $+4.8 \pm 2.9$ |
| **GCN Skip-$\alpha$** | $+10.0 \pm 6.5$ | $+3.2 \pm 7.0$ | $+2.0 \pm 5.4$ | $+3.2 \pm 4.7$ | $+5.2 \pm 5.1$ | $+6.0 \pm 6.6$ | $+3.6 \pm 3.2$ |
| **GCN Skip-PC** | $\mathbf{+15.6 \pm 3.9}$ | $\mathbf{+9.2 \pm 5.9}$ | $\mathbf{+3.6 \pm 6.0}$ | $+4.8 \pm 4.5$ | $+4.0 \pm 5.5$ | $+7.6 \pm 7.7$ | $1.6 \pm 3.2$ |
| **MLP** | $-0.4 \pm 3.2$ | $-6.8 \pm 5.3$ | $-0.4 \pm 6.5$ | $+3.2 \pm 5.5$ | $+6.8 \pm 3.6$ | $+11.2 \pm 3.7$ | $\mathbf{+11.6 \pm 2.9}$ |

Table 8: Certified ratios in [%] calculated with our exact collective certificate on **WikiCSb** for $\epsilon \in \{0.05, 0.1, 0.15, 0.2, 0.25, 0.3, 0.5\}$. As a baseline for comparison, the certified ratio of a GCN is reported. Then, for the other models, we report their *absolute change* in certified ratio compared to a GCN, i.e., their certified ratio minus the mean certified ratio of a GCN. The most robust model for a choice of $\epsilon$ is highlighted in bold.

| $\epsilon$ | 0.05 | 0.10 | 0.15 | 0.20 | 0.25 | 0.30 | 0.50 |
|---|---|---|---|---|---|---|---|
| **GCN** | $80.0 \pm 6.1$ | $71.6 \pm 4.8$ | $54.4 \pm 5.4$ | $42.0 \pm 7.6$ | $36.4 \pm 6.9$ | $31.6 \pm 9.5$ | $4.0 \pm 2.8$ |
| **SGC** | $\mathbf{+9.2 \pm 9.1}$ | $\mathbf{+2.4 \pm 15.8}$ | $-6.0 \pm 11.1$ | $-0.4 \pm 5.1$ | $\mathbf{+2.4 \pm 6.6}$ | $+0.0 \pm 13.3$ | $-2.8 \pm 1.0$ |
| **APPNP** | $+7.2 \pm 6.3$ | $-20.0 \pm 5.0$ | $-9.2 \pm 4.8$ | $+0.0 \pm 5.1$ | $-6.8 \pm 13.8$ | $-22.8 \pm 9.2$ | $-4.0 \pm 0.0$ |
| **GCN Skip-$\alpha$** | $+2.8 \pm 4.7$ | $-0.0 \pm 5.1$ | $-3.6 \pm 10.3$ | $+0.0 \pm 9.1$ | $-1.2 \pm 8.8$ | $-0.8 \pm 9.7$ | $-1.2 \pm 2.0$ |
| **GCN Skip-PC** | $+2.4 \pm 3.9$ | $+2.0 \pm 5.0$ | $\mathbf{+1.2 \pm 3.9}$ | $\mathbf{+4.0 \pm 5.7}$ | $+2.0 \pm 7.1$ | $\mathbf{+0.8 \pm 5.0}$ | $+4.8 \pm 3.0$ |
| **MLP** | $-5.6 \pm 5.6$ | $-12.0 \pm 11.6$ | $-5.2 \pm 9.7$ | $-0.8 \pm 8.8$ | $-3.2 \pm 6.5$ | $-5.6 \pm 5.4$ | $\mathbf{+8.4 \pm 3.7}$ |

Table 9: Certified ratios in [%] calculated with our exact collective certificate on **Polblogs** for $\epsilon \in \{0.05, 0.1, 0.15, 0.2, 0.25, 0.3, 0.5\}$. As a baseline for comparison, the certified ratio of a GCN is reported. Then, for the other models, we report their *absolute change* in certified ratio compared to a GCN, i.e., their certified ratio minus the mean certified ratio of a GCN. The most robust model for a choice of $\epsilon$ is highlighted in bold.

| $\epsilon$ | 0.05 | 0.10 | 0.15 | 0.20 | 0.25 | 0.30 | 0.50 |
|---|---|---|---|---|---|---|---|
| **GCN** | $73.2 \pm 14.1$ | $42.4 \pm 3.9$ | $42.0 \pm 3.6$ | $42.0 \pm 3.6$ | $42.0 \pm 3.6$ | $42.0 \pm 3.6$ | $4.4 \pm 2.9$ |
| **SGC** | $+22.4 \pm 2.0$ | $+47.2 \pm 3.9$ | $+26.4 \pm 12.9$ | $+3.2 \pm 4.5$ | $+0.4 \pm 5.3$ | $-0.4 \pm 4.5$ | $-2.8 \pm 1.5$ |
| **GCN SkipPC** | $+10.8 \pm 5.5$ | $+15.2 \pm 10.0$ | $+2.0 \pm 2.8$ | $+1.2 \pm 2.7$ | $-0.4 \pm 2.0$ | $-0.8 \pm 1.6$ | $+5.2 \pm 5.4$ |
| **GCN Skip-$\alpha$** | $-1.2 \pm 13.4$ | $+0.0 \pm 4.8$ | $+0.4 \pm 4.8$ | $+0.0 \pm 4.2$ | $+0.0 \pm 4.2$ | $+0.0 \pm 4.2$ | $+1.2 \pm 3.9$ |
| **GIN** | $+4.8 \pm 4.0$ | $+13.2 \pm 5.6$ | $+0.8 \pm 3.7$ | $-2.8 \pm 2.0$ | $-3.6 \pm 1.5$ | $-9.6 \pm 3.4$ | $+5.6 \pm 4.4$ |
| **GraphSAGE** | $+2.4 \pm 3.2$ | $+10.4 \pm 6.3$ | $+0.8 \pm 3.7$ | $-1.6 \pm 6.0$ | $-3.6 \pm 5.6$ | $-5.2 \pm 5.7$ | $+6.0 \pm 1.5$ |
| **MLP** | $-73.2 \pm 0.0$ | $-42.4 \pm 0.0$ | $-42.0 \pm 0.0$ | $-42.0 \pm 0.0$ | $-42.0 \pm 0.0$ | $-42.0 \pm 0.0$ | $-4.4 \pm 0.0$ |

Table 10: Certified ratios in [%] calculated with our exact collective certificate on **Corab** for $\epsilon \in \{0.05, 0.1, 0.15, 0.2, 0.25, 0.3, 0.5\}$. As a baseline for comparison, the certified ratio of a GCN is reported. Then, for the other models, we report their *absolute change* in certified ratio compared to a GCN, i.e., their certified ratio minus the mean certified ratio of a GCN. The most robust model for a choice of $\epsilon$ is highlighted in bold.

| $\epsilon$ | 0.05 | 0.10 | 0.15 | 0.20 | 0.25 | 0.30 | 0.50 |
|---|---|---|---|---|---|---|---|
| **GCN** | $77.6 \pm 6.5$ | $50.4 \pm 13.5$ | $33.6 \pm 3.9$ | $29.6 \pm 4.6$ | $28.8 \pm 4.7$ | $25.2 \pm 6.0$ | $11.2 \pm 2.7$ |
| **SGC** | $+6.0 \pm 2.3$ | $+0.8 \pm 11.1$ | $+2.8 \pm 5.9$ | $+2.4 \pm 4.6$ | $+1.2 \pm 5.5$ | $+2.0 \pm 5.0$ | $-0.4 \pm 3.2$ |
| **APPNP** | $-1.6 \pm 7.3$ | $-5.2 \pm 8.6$ | $-2.8 \pm 5.7$ | $-1.6 \pm 7.2$ | $-2.4 \pm 8.0$ | $-1.6 \pm 6.9$ | $-2.4 \pm 3.5$ |
| **GIN** | $-3.2 \pm 5.6$ | $-2.4 \pm 12.1$ | $+2.0 \pm 9.2$ | $-0.8 \pm 6.5$ | $-2.4 \pm 6.2$ | $-2.4 \pm 5.5$ | $-0.4 \pm 4.5$ |
| **GraphSAGE** | $+0.0 \pm 4.6$ | $-3.2 \pm 7.3$ | $+0.4 \pm 6.7$ | $+0.8 \pm 6.7$ | $-0.4 \pm 6.6$ | $+1.2 \pm 6.7$ | $+1.6 \pm 3.0$ |
| **GCN Skip-$\alpha$** | $-0.4 \pm 5.3$ | $-1.6 \pm 9.0$ | $+3.6 \pm 6.8$ | $+1.2 \pm 3.2$ | $-1.6 \pm 5.2$ | $-1.2 \pm 5.2$ | $-0.4 \pm 2.0$ |
| **GCN Skip-PC** | $-3.6 \pm 5.8$ | $+1.2 \pm 7.3$ | $+2.0 \pm 5.4$ | $+1.6 \pm 5.2$ | $-1.2 \pm 5.4$ | $-2.0 \pm 5.3$ | $-0.8 \pm 1.5$ |
| **MLP** | $-15.6 \pm 3.6$ | $-9.2 \pm 8.1$ | $+1.2 \pm 10.9$ | $+2.4 \pm 9.0$ | $-2.4 \pm 7.0$ | $-2.8 \pm 5.6$ | $+0.4 \pm 3.4$ |

Table 11: Certified ratios in [%] calculated with our exact collective certificate on **Chameleonb** for $\epsilon \in \{0.05, 0.1, 0.15, 0.2, 0.25, 0.3, 0.5\}$. As a baseline for comparison, the certified ratio of a GCN is reported. Then, for the other models, we report their *absolute change* in certified ratio compared to a GCN, i.e., their certified ratio minus the mean certified ratio of a GCN. The most robust model for a choice of $\epsilon$ is highlighted in bold.

| $\epsilon$ | 0.05 | 0.10 | 0.15 | 0.20 | 0.25 | 0.30 | 0.50 |
|---|---|---|---|---|---|---|---|
| **GCN** | $69.6 \pm 5.3$ | $52.4 \pm 11.7$ | $41.6 \pm 12.8$ | $33.2 \pm 10.4$ | $26.8 \pm 10.5$ | $22.8 \pm 10.9$ | $9.6 \pm 6.4$ |
| **SGC** | $-0.0 \pm 6.2$ | $+2.8 \pm 6.0$ | $+4.4 \pm 5.8$ | $+5.2 \pm 4.6$ | $+5.6 \pm 3.9$ | $+4.8 \pm 4.6$ | $+1.2 \pm 1.6$ |
| **APPNP** | $-2.8 \pm 14.6$ | $-3.2 \pm 9.2$ | $-8.0 \pm 7.7$ | $-7.6 \pm 6.4$ | $-6.4 \pm 3.9$ | $-8.0 \pm 5.3$ | $-4.0 \pm 3.4$ |
| **GIN** | $-26.8 \pm 8.6$ | $-28.4 \pm 10.0$ | $-23.2 \pm 8.9$ | $-18.8 \pm 8.9$ | $-16.4 \pm 7.1$ | $-13.6 \pm 6.6$ | $-5.2 \pm 3.2$ |
| **GraphSAGE** | $-5.2 \pm 4.5$ | $-8.4 \pm 12.6$ | $-6.4 \pm 11.6$ | $-2.4 \pm 11.9$ | $-1.2 \pm 11.4$ | $-0.4 \pm 11.5$ | $+0.0 \pm 8.0$ |
| **MLP** | $-29.6 \pm 22.1$ | $-38.8 \pm 4.5$ | $-29.2 \pm 4.6$ | $-21.2 \pm 4.7$ | $-16.0 \pm 4.1$ | $-12.4 \pm 4.1$ | $-6.0 \pm 2.3$ |

Table 12: Certified ratios in [%] calculated with our exact collective certificate on **CSBM** for $\epsilon \in \{0.05, 0.1, 0.15, 0.2, 0.25, 0.3, 0.5, 1\}$. As a baseline for comparison, the certified ratio of a GCN is reported. Then, for the other models, we report their *absolute change* in certified ratio compared to a GCN, i.e., their certified ratio minus the mean certified ratio of a GCN. The most robust model for a choice of $\epsilon$ is highlighted in bold.

| $\epsilon$ | 0.05 | 0.10 | 0.15 | 0.20 | 0.25 | 0.30 | 0.50 |
|---|---|---|---|---|---|---|---|
| **GCN** | $85.7 \pm 6.5$ | $67.0 \pm 9.7$ | $48.0 \pm 6.0$ | $44.7 \pm 4.3$ | $40.8 \pm 5.2$ | $34.7 \pm 7.0$ | $2.9 \pm 1.7$ |
| **SGC** | $+7.8 \pm 2.8$ | $+21.9 \pm 5.0$ | $+34.3 \pm 8.3$ | $+22.3 \pm 16.0$ | $+9.2 \pm 20.8$ | $-7.1 \pm 19.4$ | $-1.7 \pm 0.8$ |
| **APPNP** | $-2.1 \pm 7.4$ | $-6.2 \pm 7.2$ | $-1.7 \pm 3.4$ | $-0.3 \pm 2.4$ | $+1.8 \pm 1.8$ | $-0.3 \pm 6.7$ | $+1.7 \pm 1.6$ |
| **GIN** | $-3.4 \pm 8.7$ | $-2.8 \pm 13.0$ | $+2.6 \pm 9.0$ | $-5.1 \pm 7.3$ | $-9.4 \pm 5.8$ | $-10.4 \pm 7.4$ | $+2.3 \pm 2.7$ |
| **GraphSAGE** | $-0.2 \pm 4.7$ | $-2.3 \pm 8.0$ | $+0.6 \pm 4.4$ | $-3.0 \pm 2.1$ | $-3.0 \pm 3.9$ | $-3.4 \pm 9.1$ | $+1.4 \pm 2.0$ |
| **GCN Skip-$\alpha$** | $-0.1 \pm 6.4$ | $-0.9 \pm 8.0$ | $+2.1 \pm 6.6$ | $-0.3 \pm 3.7$ | $+0.6 \pm 3.9$ | $-2.7 \pm 10.0$ | $+0.8 \pm 2.2$ |
| **GCN Skip-PC** | $+4.9 \pm 3.0$ | $+15.0 \pm 6.2$ | $+20.1 \pm 9.6$ | $+7.6 \pm 12.0$ | $-2.2 \pm 13.2$ | $-5.4 \pm 13.1$ | $-0.8 \pm 1.0$ |
| **MLP** | $-9.6 \pm 2.3$ | $-16.3 \pm 4.3$ | $-4.2 \pm 3.2$ | $-1.3 \pm 3.2$ | $-4.0 \pm 4.5$ | $-5.1 \pm 7.9$ | $+6.0 \pm 2.6$ |

## H.2.2 COLLECTIVE ROBUSTNESS OF ALL ARCHITECTURES

Fig. 7 shows the certified ratio as computed with our collective certificate for all investigated architectures on different datasets.

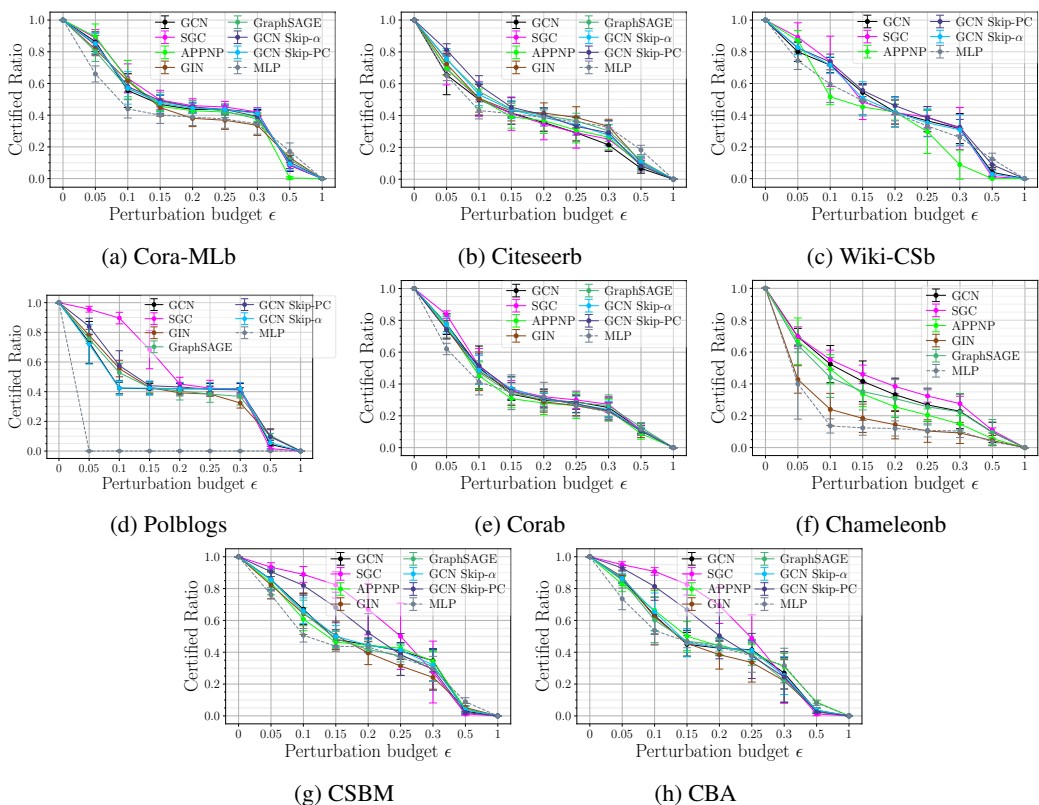

Figure 7: Certified ratio computed with our collective certificate for all investigated models.

## H.2.3 ROBUSTNESS PLATEAUING PHENONMENON

The strength of plateauing appears to depend on both the dataset and the model architecture. The Polblogs dataset shows the strongest plateauing effect out of all datasets. This indicates, as Polblogs has no features, that in a graph context, the effect is more strongly pronounced if the features carry less information compared to the structure. While for Polblogs for all architectures, for Cora-MLb for all architectures and for CSBM for many architectures, the emergence of a robustness plateau for intermediate $\epsilon$ is strikingly visible (see e.g., Fig. 7), the picture is more subtle for Citeseerb, Wiki-CSb and Chameleon. Focusing on Citeseerb, while for all architectures, the effect of increasing $\epsilon$ reduces for larger $\epsilon$, it is not immediately visible from Fig. 7b if this effect is particularly pronounced at intermediate budgets or continuously goes on until $\epsilon = 1$. Indeed, some architectures seem to show a continuous plateauing to $0$ for $\epsilon = 1$. However, if one compares the mean certified ratio difference from $\epsilon = 0.1$ to $\epsilon = 0.3$ ($\Delta_{med}$) to the one from $\epsilon = 0.3$ to $\epsilon = 0.5$ ($\Delta_{strong}$), we can find architectures showing a stronger plateauing phenomenon for intermediate $\epsilon$. Exemplary, for GIN $\Delta_{med} = 17.6\%$ compared to $\Delta_{strong} = 22.4\%$ and for MLP $\Delta_{med} = 10.4\%$ compared to $\Delta_{strong} = 14.4\%$ (also see Tab. 7). This closer study suggests that both structural and statistical properties of the data, as well as architectural design choices, jointly influence this behavior.

## H.2.4 ROBUSTNESS RANKINGS BASED ON COLLECTIVE CERTIFICATION

To compare robustness rankings for different perturbation budgets and datasets, Tab. 13 computes average ranks based on the average certified ratio computed by our collective certificate for 'weak' ($\epsilon \in (0, 0.1]$), 'intermediate' ($\epsilon \in (0.1, 0.3]$) and 'strong' ($\epsilon \in (0.3, 1)$) perturbation strengths (we

exclude $\epsilon = 1$, as all models have a certified ratio of 0). Tab. 13 shows that robustness rankings are highly data dependent as already seen in the sample-wise case, and also highly depend on the strength of the adversary.

Table 13: *Average rank* based on the average certified ratio computed using our exact collective certificate for 'weak' ($\epsilon \in (0, 0.1]$), 'intermediate' ($\epsilon \in (0.1, 0.3]$) and 'strong' ($\epsilon \in (0.3, 1)$) perturbation strengths and different datasets. The most robust model is highlighted in bold and the least robust in red. Total refers to $\epsilon \in (0, 1)$.

| | **Cora-MLb** | | | | **Citeseerb** | | | | **CSBM** | | | |
|---|---|---|---|---|---|---|---|---|---|---|---|---|
| $\epsilon$ | (0, 0.1] | (0.1, 0.3] | (0.3, 1) | total | (0, 0.1] | (0.1, 0.3] | (0.3, 1) | total | (0, 0.1] | (0.1, 0.3] | (0.3, 1) | total |
| **GCN** | 5.0 | 3.5 | 6.0 | 4.29 | 7.0 | 6.75 | 8.0 | 7.0 | 3.0 | 3.5 | 6.0 | 3.71 |
| **SGC** | 1.5 | 1.0 | 6.0 | 1.86 | 6.0 | 7.5 | 5.0 | 6.71 | 1.0 | 2.5 | 8.0 | 2.86 |
| **APPNP** | 1.5 | 5.75 | 8.0 | 4.86 | 4.5 | 6.5 | 6.0 | 5.86 | 6.5 | 3.75 | 3.0 | 4.43 |
| **GIN** | 4.5 | 7.75 | 2.0 | 6.0 | 4.0 | 1.75 | 3.0 | 2.57 | 6.5 | 6.75 | 2.0 | 6.0 |
| **GraphSAGE** | 6.5 | 4.0 | 3.0 | 4.57 | 2.5 | 2.5 | 2.0 | 2.43 | 5.0 | 5.5 | 4.0 | 5.14 |
| **GCN Skip-$\alpha$** | 4.5 | 3.25 | 5.0 | 3.86 | 2.5 | 4.0 | 4.0 | 3.57 | 4.0 | 3.5 | 5.0 | 3.86 |
| **GCN Skip-PC** | 4.5 | 3.0 | 4.0 | 3.75 | 1.0 | 3.0 | 7.0 | 3.0 | 2.0 | 3.75 | 7.0 | 3.71 |
| **MLP** | 8.0 | 7.25 | 1.0 | 6.57 | 8.0 | 3.75 | 1.0 | 4.57 | 8.0 | 6.5 | 1.0 | 6.14 |

### H.2.5 EFFECT OF DEPTH

We analyze the influence of depth in detail in this section and present ($i$) across depths and datasets, skip-connections, GCN Skip-PC and GCN Skip-$\alpha$, results in certifiable robustness that is *consistently better or as good as* the GCN. Fig. 8 demonstrates it for Cora-MLb, CSBM and CBA for $L = \{1, 2, 4\}$. ($ii$) depth, in general, decreases the certifiable robustness as observed in Fig. 9. In some cases, it is as good as $L = 1$ and only in Cora-MLb for GCN, $L = 4$ is better for small perturbations while $L = 2$ is still worse than $L = 1$.

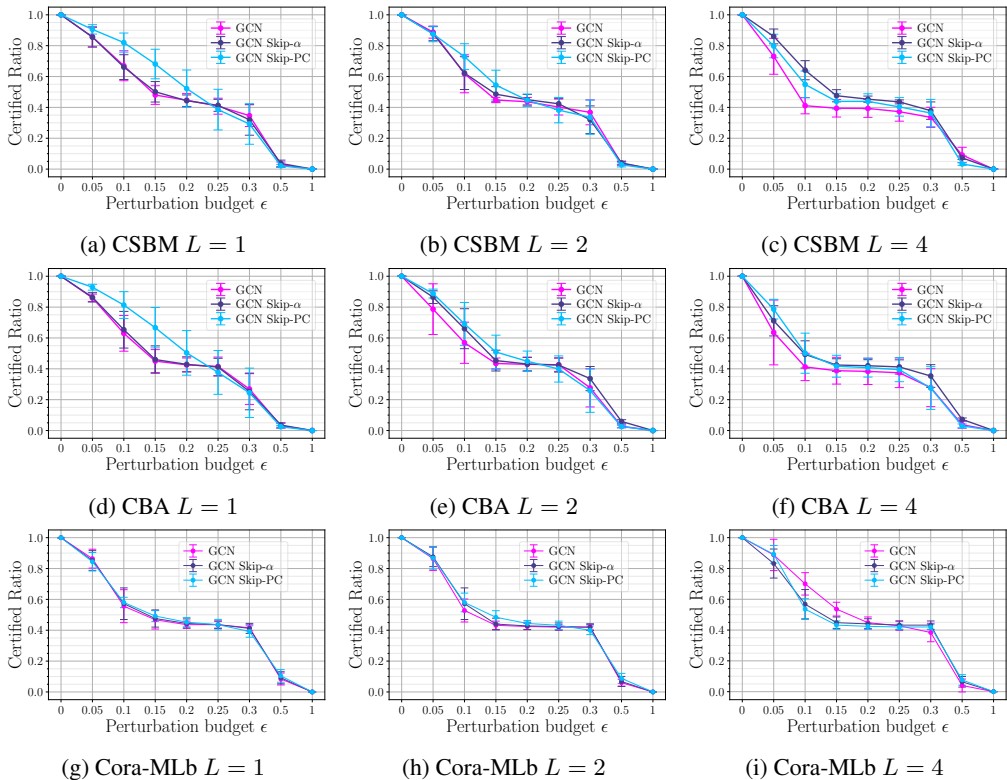

Figure 8: Effect of skip-connections showing GCN Skip-PC and GCN Skip-$\alpha$ results in certifiable robustness that is consistently better than GCN across Cora-MLb, CSBM and CBA and depths $L = \{1, 2, 4\}$.

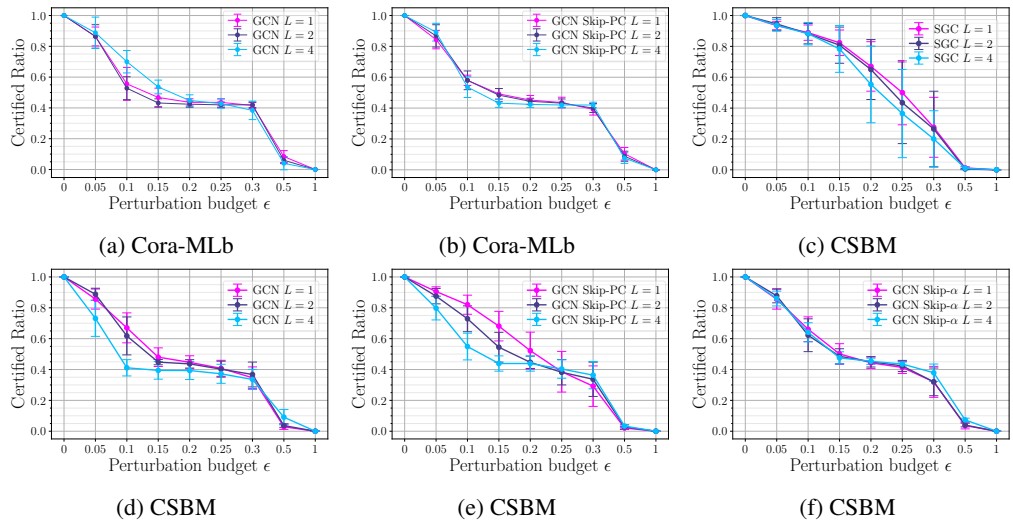

Figure 9: Effect of depth for GCN, SGC, and GCN with skip-connections showing the depth in general affects certifiable robustness negatively.

### H.2.6 EFFECT OF GRAPH CONNECTIVITY

Fig. 10 shows the increased connection density and homophily in the graphs increases certifiable robustness across GNNs such as GCN and SGC, using CSBM and CBA. Sample-wise certificates for all considered GNNs showing the same observation is demonstrated in Fig. 11. It is interesing to also note that the hierarchy of GNNs remains consistent across the settings.

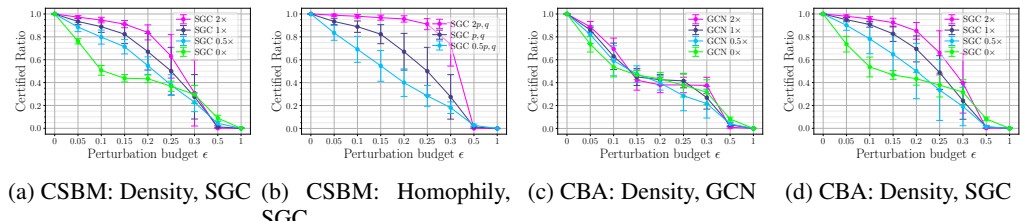

Figure 10: Effect of graph structure showing increased connection density and homophily in the graphs increases certifiable robustness.

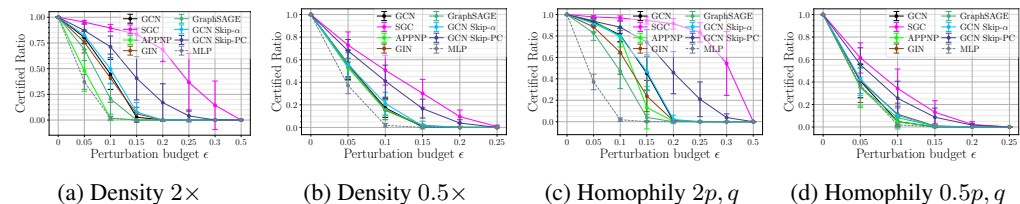

Figure 11: Sample-wise certificates for CSBM on the effect of graph structure showing increased connection density and homophily in the graphs increases certifiable robustness evaluated on CSBM.

### H.2.7 EFFECT OF GRAPH SIZE

In this section, we show that the results are consistent when the number of labeled nodes are increased to 20 nodes per class using CSBM. Fig. 12 shows the sample-wise and collective certificates showing similar behavior as the ones computed using $n = 10$. It is interesting to note that the hierarchy of GNNs observed in sample-wise certificate for $n = 20$ is the same as $n = 10$. The plateauing

phenomenon is also observed. Fig. 16 shows representative results showing the depth analysis and graph structure analysis also results in the same finding.

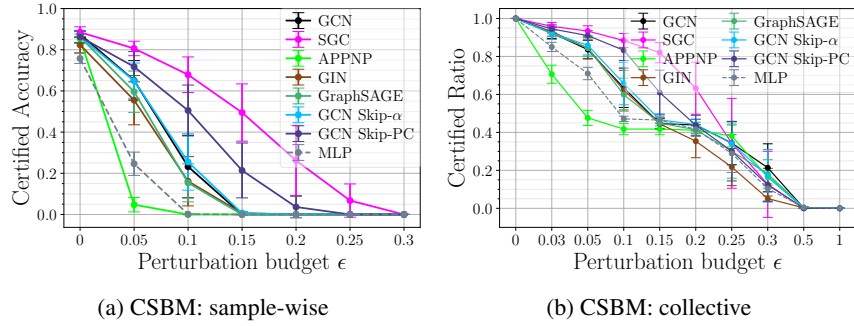

(a) CSBM: sample-wise      (b) CSBM: collective

Figure 12: The results are consistent when $n = 20$ per class is considered for CSBM. Figure showing sample-wise and collective certificates

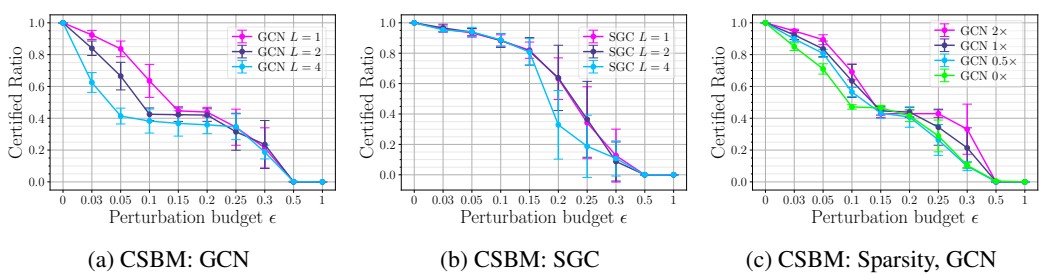

(a) CSBM: GCN      (b) CSBM: SGC      (c) CSBM: Sparsity, GCN

Figure 13: Consistency of results for a larger number of labeled nodes shown using CSBM.

### H.3   GENERALITY OF CERTIFICATES TO OTHER MODELS

In addition to MLP (a non-graph neural network architecture), we demonstrate the applicability of our certificates to other non-GNN based models such as linear kernel $XX^T$, where $X$ is the feature matrix. The collective certificate results for Cora-MLb and the random graph models CSBM and CBA is provided in Fig. 14. Our experiments demonstrate that the certificates are directly applicable to kernels and standard networks, such as fully connected and convolutional networks. Since our primary focus is on the graph node classification problem, convolutional networks were not included in this study, but their inclusion would follow the same methodology.

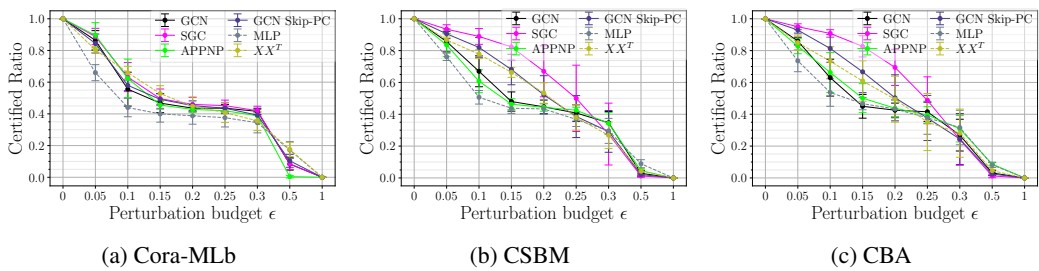

(a) Cora-MLb      (b) CSBM      (c) CBA

Figure 14: Generality of certificates to other models demonstrated using linear kernel on Cora-MLb, CSBM and CBA.

### H.4 CERTIFICATES FOR DYNAMIC GRAPHS

Our certification framework is easily adaptable to dynamic graph settings depending on the learning strategy. To demonstrate it, we consider an inductive setting where the training graph grows during inference. In Fig. 15, we provide the collective certificate results for Cora-MLb by inductively adding the test nodes to the training graph. Results are comparable to the static graph analysis.

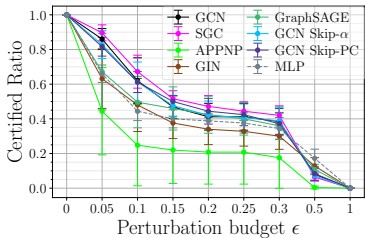

Figure 15: Certified accuracy computed with our sample-wise certificates for CBA dataset.

Furthermore, other learning methods such as aggregating temporal structural and/or feature information through summing over the temporal information (Kazemi et al., 2020) is also possible without any modification to the certificate and adapting only the adjacency and/or feature matrices in NTK computation. While we demonstrate adaptability to certain dynamic graph learning settings, we acknowledge that extending the framework to handle highly dynamic scenarios with frequent structural changes remains a promising area for future research. Incorporating temporal NTK computation or online certification methods could further enhance its applicability.

## I MULTI-CLASS EXPERIMENTAL RESULTS

We run our exact sample-wise multi-class certificate for Citeseer for selected architectures (Fig. 16a) and the inexact sample-wise variant for Cora-ML (Fig. 16b). In App. A.2 we show that our relaxed certificate provides the same or nearly as good certified accuracy as the exact certificate, while being much more scalable. Thus, for more details on the incomplete multi-class certificate including a detailed tightness discussion, we refer to App. A.2.

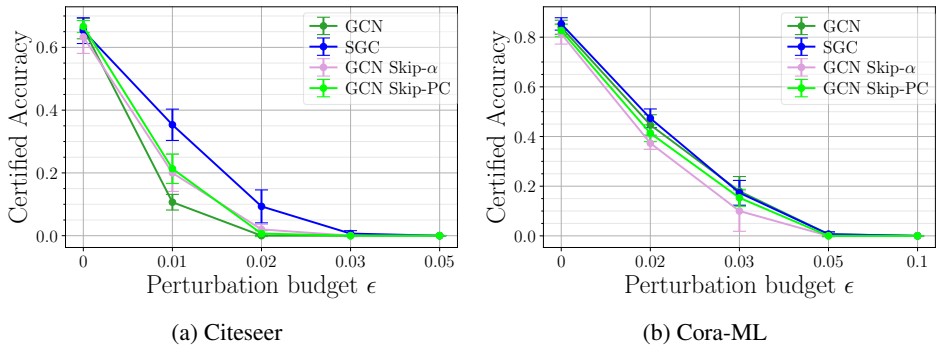

(a) Citeseer

(b) Cora-ML

Figure 16: Sample-wise certificates for multi-class datasets.

