# OpenReview forum: "Exact Certification of (Graph) Neural Networks Against Label Poisoning"
_ICLR.cc/2025/Conference — ICLR 2025 Spotlight_

### Official Review · Reviewer_5hi6 · 2024-10-23

**Soundness:** 4
**Presentation:** 3
**Contribution:** 2
**Rating:** 8
**Confidence:** 2

**Summary:**

This paper introduces exact certificates for graph neural networks against label poisoning. The author reformulate the certification problem into a Mixed Integer Linear Program (MILP) using  Neural Tangent Kernel (NTK), deriving both sample-wise and collective certificates for GNNs in node classification tasks. The method is tested on various GNNs and datasets.

**Strengths:**

- The paper is well-written and easy to follow. It introduces first exact certificates for graph neural networks against label poisoning.
- The evaluation presents insights that guide future designs and improvements in GNNs regarding label poisoning.

**Weaknesses:**

+ The NTK is data dependent, but the paper only evaluates two real-world datasets, Cora-ML and Citeseer. This raises concerns about the generalizability of the findings. It would strengthen the paper to evaluate on a broader variety of real-world datasets.
+ Given that the core technique of NTK-based MILP formulation is already established [1], the contribution seems incremental.

References:

[1] Gosch, Lukas, et al. "Provable Robustness of (Graph) Neural Networks Against Data Poisoning and Backdoor Attacks." arXiv preprint arXiv:2407.10867 (2024).

**Questions:**

In addition to the concerns mentioned in the weaknesses, some figures are not very reader-friendly, such as Figures 2c 3c. Consider adjusting the colors and legends for better clarity.

---

> ### Author Response · Authors · 2024-11-22
>
> We thank you for your positive feedback! Based on your comments, we have now significantly expanded our experimental evaluation, solidifying our results. We address your mentioned weaknesses and questions in detail below. References [A]-[D] are provided in the global response.
>
> ## W1 - Evaluate on a broader variety of real-world datasets
>
> We have now expanded our experiments to include four additional real-world graphs: Wiki-CS [A], Polblogs [B], Cora [C], and Chameleon [D] (a heterophilic dataset) - leading to a total of *six* real-world datasets investigated. We evaluate our binary class certificates on these datasets similar to Cora-ML and Citeseer and provide the results in Figure 7, and Tables 7-10, with detailed discussion in Appendix G.2.2. We observe that the results discussed in Section 4 generalize to these datasets as well. This is especially interesting, as some of these datasets have very different characteristics to Cora-ML and Citeseer, which were already included in the original submission. In particular, Wiki-CS has significantly fewer class-homogeneous neighborhoods than the common citation datasets [1], Polblogs has no node-features and is purely dependent on the graph structure, and Chameleon is a heterophilic dataset.
>
> ### W2 - Regarding the contribution
>
> We respectfully disagree that our contribution is incremental. While [1] also reformulates the bilevel problem associated to data feature poisoning by leveraging the SVM equivalence, the technical challenges and resulting contributions are fundamentally different, as outlined below:
>
> 1. **Difference in adversary:** [1] addresses the *feature poisoning* setting, whereas we focus on a different problem of *label poisoning*, which introduces distinct challenges.
> 1. **Difference in the final results:**
>     - While [1] provided only **incomplete** sample-wise certificates, we derive **exact** sample-wise certificates. Crucially, we also derive an exact **collective** certificate. Note that collective certificates are at least as important as sample-wise certificates as substantially established in our work (refer to Section 4.1). However, [1] did not provide a collective certificate.
>     - Through the exactness of our certificates, we gain **valuable insights** on architecture and data-specific properties on robustness, and **reveal** a novel phenomenon of robustness plateauing.
> 3. **Technical differences:** There are many technical differences to [1] emerging right from the first step:
>     -  In [1], the single-level reformulation obtained after employing the KKT conditions results in a **bilinear** optimization (product of two continuous variables in Eqs. 4, 5, 7). This bilinear problem is then translated to a MILP by introducing auxiliary variables with *relaxations*. In our case, the single-level reformulation using the KKT conditions results in a **multilinear** optimization, involving products of a continuous variable with two binary variables (Eq. 5), along with bilinear terms (in Eqs. 4 and 6). These distinctions make the techniques in [1] not applicable to our problem. We address these challenges by systematically introducing auxiliary variables and modeling them *exactly* using linear constraints to derive the MILP.
>     -  As [1] did not tackle collective certification, we introduce a **novel modeling strategy** to tackle this problem.
>     -  For the multi-class setting, [1] provides an incomplete sample-wise certificate based on naively applying their binary-class certificate for each class consecutively. In contrast, we provide an exact multi-class sample-wise certificate by **directly developing** a novel MILP formulation for the multi-class case.
>
> We also want to highlight that deriving exact certificates is a major advancement, as it enables deeper insights into architecture and data-specific properties influencing robustness (refer Section 4). We tried to make these differences clear in the related work Lines 173-176 and in the Introduction Lines 94-96, but have now included an extended discussion on the differences in Appendix E and referred to it in Lines 176-177.
>
> ### Q1: Adjusting Figures 2c and 3c
>
> All our figures are created with a color-blind friendly coloring scheme and provided in high-quality PDFs for clarity even upon zooming. In particular, this is also true for Figures 2c and 3c, which are created in the same manner as the other figures. However, we are happy to improve them to be more reader-friendly and would like to ask what the concrete problems are with the current coloring or legend. This would help us to address this concern more effectively.
>
>
> [1] Gosch et al. Provable Robustness of (Graph) Neural Networks Against Data Poisoning and Backdoor Attacks. Arxiv 2024.

---

> > ### Comment · Reviewer_5hi6 · 2024-11-22
> >
> > Thanks for the clarifications. I have raised my score to 8.

---

> > > ### Author Response · Authors · 2024-11-26
> > >
> > > We are happy to have addressed all your concerns and thank you for raising the score, and recommending acceptance of our paper!

---

### Official Review · Reviewer_9xfS · 2024-11-04

**Soundness:** 3
**Presentation:** 3
**Contribution:** 4
**Rating:** 8
**Confidence:** 3

**Summary:**

The paper presents a novel approach to certify Graph Neural Networks (GNNs) against label poisoning attacks by introducing exact sample-wise and collective robustness certificates. The authors leverage the Neural Tangent Kernel (NTK) to capture training dynamics, reformulating the label-flipping problem as a Mixed-Integer Linear Program (MILP). This method allows for exact robustness certification across multiple GNN architectures in node classification tasks, establishing robustness hierarchies and investigating the effect of architecture-specific choices on resilience to label flipping. A key contribution is the discovery of a plateau effect in robustness for intermediate perturbation budgets, which has implications for understanding adversarial robustness in GNNs.

**Strengths:**

**The Problem Addressed is Highly Relevant**

Poisoning attacks, of which label flipping is one variant, are highly relevant in machine learning and pose a significant security risk. While empirical defenses can sometimes weaken this effect, it is prudent to develop methods that can guarantee their absence and accurately measure their influence. The paper thus addresses a key challenge to develop more secure and trustworthy models.

**The Proposed Method is Sound Under The Infinite-Width Assumption**

The proposed method is sound under the infinite-width assumption. While this assumption does not hold in practice (see weaknesses), it gives a good theoretical basis for reasoning about model behavior. Computing exact (and not just sound but incomplete) certificates has several advantages, including a better comparison of different models’ robustness.

**Extensive Evaluation**

Considering the method's limited scalability, the evaluation is thorough, spanning several models and datasets. I find the finding of plateaus especially intriguing, which has potential for additional investigations in the future.

**Good Reproducibility**

The authors provide many implementation details in the paper and the appendix, and include their code for reproducibility.

**Weaknesses:**

**The “Exact” Certificates are Approximations**

The authors claim exact certificates (e.g., L68), which means the certificate has neither false positives nor false negatives. However, this only holds under the assumption of infinite-width models, and not for finite-width models used in practice. While I am not an expert in NTKs, my understanding is that there can be significant deviations in training behavior, especially for deeper models. I would therefore argue that the framing of the certificates should be adjusted to account for this, putting less emphasis on the certificates being “exact” for neural networks and highlighting more the fact that they are approximations. The authors acknowledge this in the conclusion; however, it should be reflected in the claims (e.g., section 1, abstract) as well.

**Limited Scalability**

The second major limitation is the method’s inherent limited scalability to small-scale models and graphs. This is common to all certification methods, but especially prevalent in exact methods. I thus don’t see this as a reason against acceptance, but it does limit the method’s practical applicability and raises questions of whether the findings hold in larger-scale settings.

**Questions:**

L132: The NTK Q_{ij} is defined as the expectation over all model initialisations. Does this mean the certificate also holds wrt. the expectation over model initialisations? If so, it would be good to highlight this, as it deviates from most work on certification which computes certificates wrt. the concrete parameter initialisation used for the particular model training.

---

> ### Author Response · Authors · 2024-11-22
>
> We thank you for the thorough and positive review and the support for acceptance. We incorporated your raised points into the updated draft and thereby, improved upon our initial submitted version. We address the weaknesses and questions in detail below.
>
> > The “Exact” Certificates are Approximations
>
> It is correct that finite-width networks can show significant deviations in training behavior. We did adjust the framing of our method, but before we go into details about the changes, we want to clarify that the exactness of the certificates transports to finite-width networks with high probability, *if they have sufficient width*. Concretely, the output difference between an infinite-width network and a finite-width network with width $w$ is bounded by $\mathcal{O}(\frac{\ln w}{\sqrt{w}})$ with probability $p=1-\exp(-\Omega(w))$. This means that the output difference approaches $0$ with high probability for increasing $w$ and given a sufficiently large $w^*$, the output difference will be too small to change the worst-case prediction as calculated by our certificate (e.g., using the MILP from Thm. 1). Thus, given finite but sufficient width, our certificate is still exact, but the exact guarantee only holds with probability $p$ as the bound only holds with this probability, and with probability $1-p$ no (exact or inexact) statement can be made. Conceptually, this is similar to randomized smoothing [1] where the (inexact) guarantee holds with a certain probability $p'$ and doesn't hold with probability $1-p'$.
>
> We did not intend to claim exact certification of finite but not sufficiently wide networks, and we tried to make this clear in the original draft (Lines 81, 89-90, and 109 as well as in the "On Finite Width Certification" discussion in conclusion). However, we agree that this should be made more clear and the framing of the method adjusted to avoid any potential misunderstandings. Thus, we made the following clarifying changes to the draft:
>
> * We updated the abstract (Line 26) and introduction (Line 81) to clearly state *sufficient width*.
> * We updated our claim $(i)$ in the introduction to explicitly state that our certificate applies to infinite-width NNs and with high probability for finite but sufficiently wide networks (Lines 100-101).
> * We moved the technical discussion on finite-width certification from the conclusion to the preliminaries (see Lines 150-156) and further extended it with a discussion on the probability with which the certificate holds for finite-width networks.
>
> We are confident to now clearly convey when our certificate is exact throughout the manuscript and are open to welcome any additional input or suggestions.
>
> > L132: The NTK Q_{ij} is defined as the expectation over all model initialisations. Does this mean the certificate also holds wrt. the expectation over model initialisations?
>
> Thank you for this insightful observation! You are correct that the certificates hold with respect to the expectation over model initializations. As a result, our certificates can be understood as providing guarantees at the population level of the parameters, offering a perspective that differs from most certification methods, which typically focus on guarantees for a specific, fixed initialization of the model. We have now added a paragraph to the conclusion (Lines 525–529) to highlight this distinction.
>
>
>
> [1] Cohen et al. "Certified Adversarial Robustness via Randomized Smoothing", ICML 2019

---

> ### Comment · Reviewer_9xfS · 2024-11-22
>
> Thank you for addressing my comments! I agree with your point that the certificates are exact with probability $p$ - I should have phrased this better. I appreciate the added clarifications, particularly regarding the qualitative bounds early in the paper.
>
> Is it possible to explicitly compute or bound the probability $p$ for a given model? If so, it would be valuable to include this for the models used in the experiments.

---

> ### Author Response · Authors · 2024-11-26
>
> Thank you for the response! In the following, we address your follow-up question.
>
> It is possible to derive a bound for the probability $p$ (and the output difference) that more explicitly takes the model architecture into account. Thus, we now include the bounds on the probability and output deviation to include the depth, width, and activation functions used in a model in Appendix E and refer to it in Lines 155-156 (marked red). To obtain this, we follow the derivations in [1,2]. Concretely, we consider a graph neural network with depth $L$, width $w$, employing activation functions with Lipschitz constant $\rho$, and trained using the regularized Hinge loss with $C$ as the regularization constant. Let the network parameters $W$ during training move within a fixed radius $R>0$ to initialization $W_{init}$, i.e. $\\{W |\ ||W-W_{init}|| \leq R\\}$. Then, we get the bound for the output difference between an infinite-width and a finite-width network as $\mathcal{O}(\frac{R^{3L+1}\rho \ln w}{C w})$ with probability $p=1-L\exp(-\Omega(w))$.  However, this theoretical bound is not directly computable unless constants in the derivation are preserved and applied to specific inputs. Unfortunately, the literature on the NTK so far is mainly concerned with providing convergence statements in big-O notation and not with calculating the individually involved constants. While we believe it is possible to derive such constants, this is a complex and intricate theoretical task that we find interesting, but outside of our scope. We now include a discussion on the computability of these bounds in Appendix E Lines 1005-1009.
>
> [1] Liu et al. On the linearity of large non-linear models: when and why the tangent kernel is constant. NeurIPS 2020
>
> [2] Chen et al. On the equivalence between neural network and support vector machine. NeurIPS 2021

---

> > ### Comment · Reviewer_9xfS · 2024-11-26
> >
> > Thank you for the follow-up and more precise error terms. While not being able to compute a concrete bound limits the method's practical applicability, the paper gives interesting insights into the expected training behavior and the influence of different model parameters. I continue to support the acceptance of the paper.

---

> > > ### Author Response · Authors · 2024-12-02
> > >
> > > Thank you for the positive response and again for your support to accept our paper!

---

### Official Review · Reviewer_ktGU · 2024-11-09

**Soundness:** 3
**Presentation:** 3
**Contribution:** 3
**Rating:** 8
**Confidence:** 2

**Summary:**

The paper "Exact Certification of Graph Neural Networks Against Label Poisoning" introduces the first exact certification method, for deriving complete sample-wise and collective robustness certificates for Graph Neural Networks, against adversarial label flipping.
Using the Neural Tangent Kernel (NTK), the authors leverage the dynamics of wide networks in order to recast the bilevel optimization problem of label flipping, as a Mixed-Integer Linear Program (MILP). In this work, the authors establish hierarchies of GNNs on different benchmark graphs, and quantify the effect of architectural options to the worst-case robustness flipping.

**Strengths:**

1.   The authors introduce a novel exact robustness certification method for label flipping attacks on neural networks, particularly Graph Neural Networks (GNNs). This work is impactful given that exact certification for poisoning attacks is generally unsolved for GNNs​.
2. By leveraging the Neural Tangent Kernel (NTK), the paper rigorously reformulates the robustness certification problem as a Mixed-Integer Linear Program (MILP). The authors were able to extend previous work [1], deriving complete certificates, and collective certification.
3. The approach offers wide applicability, as it can extend beyond GNNs to general neural networks, offering a framework that could potentially influence robustness research across various domains in machine learning​.
4. The authors identified a novel phenomenon of robustness plateauing for intermediate perturbations.


[1]. Lukas Gosch, Mahalakshmi Sabanayagam, Debarghya Ghoshdastidar, and Stephan G¨unnemann. Provable robustness of (graph) neural networks against data poisoning and backdoor attacks. arXiv preprint arXiv:2407.10867, 2024.

**Weaknesses:**

1. The high complexity of Mixed-Integer Linear Program (MILP)  could introduce computational overhead, which may be prohibitive for real-time or large-scale applications.
2. Additionally, the certification method is largely tested on synthetic datasets. Although effective on these datasets, it’s unclear how well it would generalize to real-world graphs.
3. Finally, the paper focuses solely on exact certification. The authors are encouraged to consider trade-offs between exactness and scalability.

**Questions:**

1. Scalability and Complexity: Given the computational complexity of your MILP-based certification, have you considered or tested any strategies to approximate or simplify the certification process for larger datasets? Would relaxing the exactness of the certification improve scalability while retaining useful guarantees?

2. Dataset Dependence: Your results highlight that robustness hierarchies among GNN architectures vary across datasets. Can you elaborate on why certain architectures may perform better on specific datasets? What dataset characteristics influence these hierarchies?

3. Intermediate Perturbation Plateau: You observed a plateau in robustness for intermediate perturbation budgets across several GNN architectures and datasets. Do the authors have any hypotheses about the causes of this phenomenon, and do you plan further analysis to understand it more deeply?

4. Extension Beyond GNNs: While the method can theoretically be applied to other types of neural networks, have you tested it on non-GNN architectures? If so, how did the performance and scalability compare?

5. Unaddressed Adaptability to Dynamic Graphs: Real-world graphs often change over time, yet the certification process assumes a static graph. Can your methodology adapt to provide certification for dynamic or evolving graphs?

---

> ### Author Response · Authors · 2024-11-22
> **Response to W1-W3, Q1-Q2**
>
> We thank you for your thoughtful and positive review! Based on your questions and concerns, we could significantly improve upon our initial submission. We address the mentioned weaknesses and questions in detail below. References [A]-[D] are provided in the global response.
>
>
> ### W2 - Method largely tested on synthetic data and generalization to real-world graphs.
>
> We have now expanded our experiments to include four additional real-world graphs: WikiCS [A], Polblogs [B], Cora [C], and Chameleon [D] (a heterophilic dataset). We evaluate our binary class certificates on these datasets similar to Cora-ML and Citeseer and provide the results in Figure 7, and Tables 7-10, with a detailed discussion in Appendix G.2.2. We observe that the results discussed in Section 4 generalize to these datasets as well.
>
> ### W1, W3 & Q1 - Trading exactness with scalability
>
> Indeed, we do consider and test a strategy to trade exactness with scalability discussed in Appendix A, Lines 764-772. In particular, we discuss how to obtain a non-exact multi-class certificate by leveraging the individual binary-class certificates instead of solving the more complex exact multi-class certificate. In Figure 16b (App. H) we show that this strategy yields useful guarantees on Cora-ML.
>
> While in our work, the focus lies on exact certification, we are also convinced that trading-off exactness with scalability is a fruitful way for future work to tackle the important problem of scaling. Thus, we want to note that we explicitly mentioned this in the "Scalability" paragraph in the Conclusion (see Line 539).
>
> ### Q2 - Discussion on data-dependent robustness hierarchies
>
>
> These are very interesting questions that we have also asked ourselves. While we cannot give the ultimate answer to why a certain architecture performs better than another, we conducted several experiments to understand how certain dataset characteristics influence robustness. We explore the influence of graph size in our original draft (see Figure 12) as well as the effect of graph density and homophily on collective robustness in Figures 5c and 5d. In the updated draft, we have extended this analysis to multiple GNNs as well as to sample-wise certification (see Figure 11 in Appendix G.2.6). Interestingly, our findings reveal consistent robustness hierarchies across these varying settings. However, our results also show that robustness properties strongly depend on underlying structural graph properties and class information.
>
> We hope that these results inspire future work on a more deeper understanding of the "why". In this context, we see our certificate as a tool to uncover such robustness hierarchies and, thereby, making them accessible for further studies.

---

> > ### Author Response · Authors · 2024-11-22
> > **Response to Q3-Q5**
> >
> > ### Q3 - Hypotheses on the plateauing of robustness
> >
> >
> > Indeed, we are very curious about this phenomenon and are actively thinking about how to further understand it. The new results on additional real datasets such as Wiki-CS, Cora, Polblogs, and Chameleon further confirm and strengthen the universality of this observation (Figure 7). Below, we elaborate on our hypotheses:
> >
> > 1. **Hypothesis 1: Universality of the plateauing phenomenon for NNs.**
> > We observed that plateauing occurs consistently for NNs across all datasets explored, including the newly added datasets. Most curiously, with our newly added results, we do not observe the phenomenon for an SVM with a linear kernel, even not on Cora-MLb, where all GNNs (including the MLP) show the phenomenon. Thus, it could be that the plateau is a unique phenomenon for neural networks (or complex enough classifiers). We further want to note that the behavior is *not* universal across all architectures for every dataset. This indicates that the plateauing phenomenon may arise from a combination of dataset-specific characteristics with architecture-specific properties (see Hypothesis 3).
> > 2. **Hypothesis 2: Influence of graph structure.**
> > Initially, we hypothesized that the graph structure (and homophily/heterophily) might fundamentally drive the plateauing effect. However, similar behavior was observed in MLPs, which do not utilize graph structure, effectively ruling out this hypothesis. However, the results on the newly added Polblogs dataset show the strongest plateauing effect out of all datasets. This indicates, as Polblogs has no features, that in a graph context, the effect is more strongly pronounced if the features carry less information compared to the structure.
> > 3. **Hypothesis 3: Interplay between data and architecture.**
> > The strength of plateauing appears to depend on both the dataset and the model architecture. Exemplary, the phenomenon is pronounced in Cora-MLb and Polblogs (near-perfect plateauing) but weaker in WikiCS and Chameleon. This suggests that both structural and statistical properties of the data, as well as architectural design choices, jointly influence this behavior.
> >
> > To further understanding of this phenomenon, we think it is important for future work to develop a formal understanding of the plateauing effect, potentially focusing on how architecture (e.g., graph convolution, activation functions, hidden layers) and data properties (e.g., homophily, graph density, node centrality) influence robustness under varying perturbation budgets.
> >
> > ### Q4 - Extensions Beyond GNNs
> >
> > In addition to an MLP i.e., a classical feed-forward (non-graph) neural network, which we used as a baseline model in most of our experiments (see e.g., Figure 3), we now provide experimental results for an SVM with a linear kernel $XX^T$, where $X$ is the feature matrix (see results in Figure 14 with discussion in Appendix G.3). While our method is directly applicable to other standard methods such as convolutional networks used for image classification or different kernalized SVM's, we did not include them in this study as our primary focus is on graph node classification. While the performance for non-GNN's varies depending on the specific model and problem setting, our certification method provides useful guarantees for these models. Furthermore, the scalability of our approach for these models remains comparable to that of using GNNs. This is because the computational cost is primarily dictated by the MILP formulation, which scales with the number of labeled samples rather than the type of neural network.
> >
> > ### Q5 - Adaptations to Dynamic Graphs
> >
> > Our certification framework can be adapted to dynamic graph settings depending on the learning strategy. To demonstrate it, we consider an inductive setting where the graph grows at test time. In Figure 15 of Appendix G.4, we provide the collective certificate results for Cora-MLb by inductively adding the test nodes to the training graph. Results are comparable to the static graph analysis. Furthermore, other learning methods such as aggregating temporal structural and/or feature information through summing over the temporal information [1] is also possible without any modification to the certificate and requires only adapting the adjacency and/or feature matrices in the NTK computation.
> >
> > While we demonstrate adaptability to certain dynamic graph learning settings, we acknowledge that extending the framework to handle highly dynamic scenarios with frequent structural changes remains an open problem and area for future research.
> >
> > [1] Kazemi et al. Representation learning for dynamic graphs: A survey. JMLR 2020.

---

> > > ### Comment · Reviewer_ktGU · 2024-11-22
> > > **Official Comment by Reviewer ktGU**
> > >
> > > I would like to thank the authors for their thorough responses and their work to improve the paper. Overall, I agree to increase my score to 8.

---

> > > > ### Author Response · Authors · 2024-11-26
> > > >
> > > > We are happy to have addressed all your concerns, and thank you for raising the score and recommending acceptance of our paper!

---

### Official Review · Reviewer_QNjQ · 2024-11-10

**Soundness:** 3
**Presentation:** 3
**Contribution:** 2
**Rating:** 6
**Confidence:** 3

**Summary:**

This work investigates the exact robustness certification for GNN against label poisoning attack. They use the NTK to approximate the wide GNN, and reformulate the bilevel optimization problem into a MILP. The authors believe that their framework can be applied to other GNNs through its NTK. Finally, they conduct experiments to show their certification for various GNNs over several datasets.

**Strengths:**

1. The robustness certification is a vital problem for GNN and other NNs. It is hard to obtain an exact robustness certification.
2. The author proposes a novel framework for robustness certification against label poisoning, and their main idea is to approximate the model by its NTK. It seems to be easy to apply this framework to other models.
3. The computational complexity of solving the MILP is not very large. The authors conduct experiments on several datasets and GNNs to show its empirical performance.

**Weaknesses:**

1. My main concern is about the approximation error when using NTK to approximate GNN. The approximation error exists unless the model is infinitely wide, especially when we use a pooling layer. However, the authors aim to obtain exact certificates, but the approximation error is not considered in this work.
2. A limitation of this framework is that it requires the width of the model to be sufficiently large, and it cannot be used for narrow NNs.
3. An important recent work is [a], which studies the robustness certification of GNN against data poisoning. I find that the techniques and framework in this work are very similar to [a], which limits the contribution of this work.
4. The authors only show the computational complexity of solving the MILP. However, since it approximates GNN by its NTK, it is necessary to show the computational complexity of computing the NTK of GNN.

[a] Lukas Gosch, Mahalakshmi Sabanayagam, Debarghya Ghoshdastidar, Stephan Günnemann. Provable Robustness of (Graph) Neural Networks Against Data Poisoning and Backdoor Attacks. Arxiv 2024.

**Questions:**

1. When considering the approximation error in this framework, is this method an exact certification method? (especially when the width of GNN is not sufficiently wide).
2. How to evaluate the tightness of the certifies computed by this method?
3. What is the total computational complexity of the proposed method?
4. What is the novelty of the techniques in this work?

---

> ### Author Response · Authors · 2024-11-22
> **Response to W1, Q1-Q3**
>
> We thank you for the constructive feedback! Based on your feedback, we improved our paper and we address all your raised weaknesses and questions below.
>
> ### W1 - Clarification of the approximation error due to NTK
>
> We kindly disagree that we do not consider the approximation error. We state that our certificates only hold for sufficiently wide networks in the introduction (lines 81, 89-90, and 109) and discuss this in detail in conclusion (see "On Finite Width Certification" Lines 506-512 in the original draft). However, we do agree that this should be made more clear to avoid any potential misunderstandings. Thus, we made several clarifying changes to the draft:
>
> *  We updated the abstract (Line 26).
> * We updated our claim $(i)$ in the introduction to explicitly state that our certificate applies to infinite-width NNs and with high probability for finite but sufficiently wide networks (Lines 100-101).
> * We moved the technical discussion on finite-width certification from the conclusion to the preliminaries (see Lines 150-156) and further extended it (see answer to Q1 & Q2 below for details).
>
> We hope this fact is now clearly conveyed throughout the manuscript and we welcome additional feedback or specific suggestions that could further improve the discussion.
>
>
> ### Q1 & Q2 - Tightness & When considering the approximation error in this framework, is this method an exact certification method?
>
> If the width of a network is not sufficiently wide, our certificate does not apply and hence, can't be used to provide an exact or inexact guarantee. However, if the width of a network is sufficiently wide, the certificate's guarantee holds with high probability, and if it holds, it is exact (tight).
>
> To clarify, the crucial fact here is that our method does not underestimate the robustness of the studied model (like an inexact certificate would). In particular, if the model is not infinitely-wide but has finite but sufficient width $w$, our method provides an exact guarantee with a certain probability $p$ and does not apply (neither providing an exact or inexact guarantee) with probability $1-p$. The probability $p$ can be derived to be $p=1-\exp^{-\Omega(w)}$ [1], which we now include in our discussion on certifying finite-width networks (Lines 150-156). Conceptually, this is similar to randomized smoothing [2], which provides an (inexact) guarantee that holds with a probability $p'$ and with probability $1-p'$ does not hold at all. Note that $p$ approaches $1$ exponentially as a function of the width $w$ and, thus, is a high-probability statement.
>
>
> [1] Liu et al. "On the linearity of large non-linear models: when and why the tangent kernel is constant", NeurIPS 2020
>
> [2] Cohen et al. "Certified Adversarial Robustness via Randomized Smoothing", ICML 2019

---

> ### Author Response · Authors · 2024-11-22
> **Response to W3-W4, Q3-Q4**
>
> ### W3 and Q4 - Discussion on technical novelty
>
> While [1] reformulates the bilevel problem associated to data feature poisoning by also leveraging the SVM equivalence, the technical challenges and resulting contributions are fundamentally different, as outlined below:
>
> 1. **Difference in adversary:** [1] addresses the *feature poisoning* setting, whereas we focus on a different problem of *label poisoning*, which introduces distinct challenges.
> 1. **Difference in the final results:**
>     - While [1] provided only **incomplete** sample-wise certificates, we derive **exact** sample-wise certificates. Crucially, we also derive an exact **collective** certificate. Note that collective certificates are as important as sample-wise certificates as substantially established in our work (refer to Section 4.1). However, [1] did not provide a collective certificate.
>     - Through the exactness of the certificates, we gain **valuable insights** on architecture and data-specific properties on robustness, and **reveal** a novel phenomenon of robustness plateauing.
> 3. **Technical differences:** The technical differences emerge right from the first step:
>     -  In [1], the single-level reformulation obtained after employing the KKT conditions results in a **bilinear** optimization (product of two continuous variables in Eqs. 4, 5, 7). This bilinear problem is then translated to a MILP by introducing auxiliary variables with *relaxations*. In our case, the single-level reformulation using the KKT conditions results in a **multilinear** optimization, involving products of a continuous variable with two binary variables (Eq. 5), along with bilinear terms (in Eqs. 4 and 6). These distinctions make the techniques in [1] not applicable to our problem. We address these challenges by systematically introducing auxiliary variables and modeling them *exactly* using linear constraints to derive the MILP.
>     -  As [1] did not tackle collective certification, we introduce a **novel modeling strategy** to tackle this problem.
>     -  For the multi-class setting, [1] provides an incomplete sample-wise certificate based on naively applying their binary-class certificate for each class consecutively. In contrast, we provide an exact multi-class sample-wise certificate by **directly developing** a novel MILP formulation for the multi-class case.
>
> We highlight that deriving exact certificates is a major advancement, as it enables deeper insights into the architecture and data-specific properties influencing robustness (refer Section 4). We included this discussion now in Appendix E and referred to it in prior works discussion in Lines 176-177.
>
> ### W4 and Q3 - Clarification on the computational complexity
>
> The computation complexity of the fixed quantities in our final MILP certificates are:
> - NTK for GNNs $Q$ is computed between all pairs of graph nodes involving simple products and sums, thus leading to $\mathcal{O}(m^2)$ complexity where $m$ is the number of labeled nodes.
> - Positive constants $M_u$ and $M_v$ are $\mathcal{O}(m)$
>
> While these inputs to MILP contribute polynomial complexity, the overall computation of the certificate is dominated by the MILP solution process, which is NP-hard with exponential complexity determined primarily by the integer variables in the problem. We note that the original poisoning objective in Eq. 3 is also NP-hard, while the MILP reformulation makes the problem tractable. To address the reviewer’s concern, we have now included this discussion in lines 270–275 of the updated manuscript.
>
> [1] Lukas Gosch, Mahalakshmi Sabanayagam, Debarghya Ghoshdastidar, Stephan Günnemann. Provable Robustness of (Graph) Neural Networks Against Data Poisoning and Backdoor Attacks. Arxiv 2024.

---

> > ### Comment · Reviewer_QNjQ · 2024-11-25
> >
> > Thanks for the response from authors. I have read the response and comments from other reviewers. Here're my remaining concerns:
> >
> > 1. The authors update the discussion about the exact guarantee for finite-width networks, and they say that "the output difference to the SVM is bounded by $O(\frac{\ln w}{\sqrt{w}})$ with probability $1 - \exp(-\Omega(w))$", and their method "provides an exact guarantee with a certain probability $p$ and does not apply (neither providing an exact or inexact guarantee) with probability $1-p$". But the bound is still not strictly exact due to the apporimation error, although the difference of output is bounded. **The certificate for the GNN is not exact but an estimation of exact certificate.** Reviewer 9xfS also points this out.
> >
> > 2. The authors say that "while [1] provided only incomplete sample-wise certificates, we derive exact sample-wise certificates", but I think the authors should explain this more clearly. It would be more convincing to me to show **why [1] provided only incomplete sample-wise certificates, but this work derives exact sample-wise certificates.**
> >
> > To summary, I keep my rating for the reasons given above. I hope the authors could kindly resolve these problems.

---

> ### Author Response · Authors · 2024-11-26
> **Response to Concern 2**
>
> We thank you for the detailed response and for clarifying your remaining concerns. Below, we address these concerns in detail, first by addressing Concern 2 and then, following up with Concern 1.
>
>
> ### Concerning 2
>
> The feature certificate provided in [1] is inherently incomplete due to the following: The bilevel feature certification problem for infinite-width networks is stated in their Eq. 4 [1]. This is reformulated into a single-level problem denoted $P_3$. Subsequently, the authors **relax** $P_3$ to make it tractable, thus deriving an **incomplete** certificate. This is explicitly written by the authors in their "Discussion and related work" - Section 5 (Page 9) under 'How tight is QPCert?' as well as in the last sentence on Page 5 in the methods section. The technical reason for this is that for feature certification, the NTK matrix $Q$ is itself a variable, as it is a function of the feature matrix $X$ over which [1] optimizes. As [1] also optimizes over the dual variables $\alpha$, the terms $\alpha_i Q_{ij}$ arising in their optimization problem $P_3$ (see their Eq. 4 & 5) are multiplications of two continuous variables which can't be exactly modeled linearly (see also the last paragraph in Section 5 in [1], where the relaxation is introduced).
>
> This contrasts with our work, where we optimize over the labels $y$ (which are constant in [1]) instead of the features leading to $Q_{ij}$ staying constant in our label certification problem. Thus, the arising terms (i) $y_i \alpha_i Q_{ij}$ and (ii) $y_i y_j \alpha_i Q_{ij}$ in our optimization problem (see Section 3.1) are multiplications of a discrete ($y_i$) with a continuous variable ($\alpha_i$) in case (i) and a multiplication of two discrete ($y_i,y_j$) with one continuous variable ($\alpha_i$) in case (ii). In Section 3.1, we introduce techniques to model these multilinear terms exactly allowing us, in conjunction with other modelling techniques we introduce in Section 3.1, to convert the bilevel label certification problem for kernelized SVM's and infinite-width networks (as provided in our Eq. 4) into a mixed-integer linear problem **in an exact way**. Thus, in contrast to [1], we *do not employ a relaxation* to solve the arising certification problem leading to an exact certificate. In a similar way, our multi-class certificate (Appendix A) and collective certificate (Section 3.2) are *exact*, whereas [1] proposes a relaxed multi-class certificate through a simple application of their incomplete sample-wise certificate in their Appendix E, and they do not provide a collective certificate.
>
> We now updated our technical comparison to [1] in Appendix F to make this point more clear (see Lines 1021/1022 and Lines 1026-1028 marked in red).
>
> [1] Lukas Gosch, Mahalakshmi Sabanayagam, Debarghya Ghoshdastidar, Stephan Günnemann. Provable Robustness of (Graph) Neural Networks Against Data Poisoning and Backdoor Attacks. Arxiv 2024

---

> ### Author Response · Authors · 2024-11-26
> **Response to Concern 1**
>
> ### Concerning 1
> To avoid any misunderstanding, we absolutely agree that in the finite-width case, there is an approximation error that is bounded by $\mathcal{O}(\frac{\ln w}{w})$, where $w$ is the width of the network. Important to recognize now is that this means that for an increasing $w$, the approximation error approaches $0$ (as $\frac{\ln w}{w} \rightarrow 0$ for $w \rightarrow \infty$). Thus, there exists a certain finite but *sufficiently large* width $w'$, at which the approximation error is small enough, so that it can't change the worst-case prediction as calculated for an infinite-width network.
>
> **Example.** Assume a binary classification task and that we are interested in certifying a node $t$. Further, assume our sample-wise certificate for an infinite-width GNN outputs that node $t$ is certifiably robust (or unrobust) with a worst-case prediction score $p_t^*$. Now, as the approximation error approaches $0$ for increasing $w$, we know that there exists (a sufficiently large) width $w'$, for which the approximation error will be smaller than $p_t^*$ and thus, can't change the sign of the prediction $p_t^*$. Then, the certificate will still be exact for all finite-width networks that have widths $\ge w'$. However, as the bound only holds with probability $p=1-\exp(-\Omega(w))$, the certificate is exact for such sufficiently wide networks with probability $p$ and can't be applied (i.e., does not provide any guarantee) with probability $1-p$. This example can be generalized to certify an arbitrary collection of nodes by considering the closest worst-case prediction to the decision boundary as given by our certificate.
>
> Thus, we agree that if one does *not* assume sufficient width, there is an approximation error that would render the certificate inexact. In particular, one would obtain an inexact certificate with probability $p$ and no guarantee at all with probability $1-p$. However, **given the sufficient width assumption**, the infinite-width certificate transports to an exact certificate for a sufficiently wide but finite network with probability $p$ as *agreed by Reviewer 9xfS* in their response to our rebuttal (see Reviewer 9xfS's Official Comment).
>
> **Paper Updates.** We did not intend to claim exact certification of finite but not sufficiently wide networks. This is the reason why we, in the revised paper, made more clear that our exact certificate is only applicable for infinite-width or finite but sufficiently wide networks (see the changes in the abstract Line 26 and Introduction Lines 81/82 and 100/101). Note that we also explicitly write this in the updated discussion on finite-width networks (Lines 150-156) cited by you, where we state *"As a result, **given sufficient width**, a guarantee for the infinite-width case will translate to a high-probability guarantee for the finite-width case."* However, we have now uploaded a new draft, where we explicitly state (see Lines 157-159 marked red) that if a finite-width network does not have sufficient width, the certificate would become incomplete. We are happy to provide more adaptations if you think this fact is still not conveyed appropriately.
>
> *Note.* To employ our certificate in an incomplete way to finite-width networks would require to explicitly compute the approximation error bounded by $\mathcal{O}(\frac{\ln w}{w})$. For this, one would need knowledge of the concrete constants involved in the bound. Unfortunately, the literature on the NTK so far is mainly concerned with providing convergence statements in big-O notation and not with calculating the individually involved constants. Calculating these is a complex and intricate theoretical task that we find interesting, but outside of our scope, and we now include a discussion on this in App. E Lines 1005-1009.
>
> ---
>
> We hope we have addressed your remaining two concerns and are happy to provide further clarifications if needed.

---

> > ### Comment · Reviewer_QNjQ · 2024-11-28
> >
> > Thanks for the further response. I acknowledge the techniques used in one of the subproblems is different to [1], which relies on the specific structure of this problem. Additionally, I think the statement that "one would obtain an inexact certificate with probability $p$ and no guarantee at all with probability $1-p$ ... with the sufficient width assumption" seems strange and not precise. The certificate of SVM is exact, but the certificate of GNN obtained by the proposed method is just asymptotically exact. The authors should clarify this in the revisions. I suggest that the authors describe this as an asymptotic result. It could be stated as "the proposed method obtains an asymptotically exact certificate as the width $w$ approaches infinity". I will raise my rating to 6 after the discussion if the clarification is clear in the revisions.

---

> > > ### Author Response · Authors · 2024-11-28
> > >
> > > Thank you for the follow-up response and suggestion! We now clarify this as proposed in your response in the revised manuscript in Lines 156-158 (marked red) where we discuss the finite-width certification and Lines 504-506 (marked red) in the conclusion.
> > >
> > > Concretely, in the finite-width certification discussion, we added the following clarifying sentence:
> > > > As a result, a certificate obtained through the SVM equivalence represents an asymptotically exact certificate as the width $w$ approaches infinity.
> > >
> > > and in the conclusion, when we discuss our proposed method, we added the following clarifying remark:
> > > > In particular, the proposed method obtains an asymptotically exact certificate as the width approaches infinity.
> > >
> > > Thereby, we hope we clarified this now in our revised paper. If you feel it could still be addressed better, we are open to further suggestions.

---

> > > > ### Comment · Reviewer_QNjQ · 2024-11-29
> > > >
> > > > Thanks for the response. I have no further concerns, and decide to raise my rating.

---

> ### Author Response · Authors · 2024-12-02
>
> We want to thank you for the positive response and for raising the score!

---

### Author Response · Authors · 2024-11-22
**Global response**

We thank the reviewers for their helpful feedback and insightful comments and questions. Based on it, we added several new and interesting experimental results and discussions, solidifying our results. The revised version of our paper includes all the updates indicated in blue. Line numbers in the rebuttal refer to the revised paper if not indicated otherwise.

Specifically, we added the following **new experiments**:

- We added experiments on **four** additional **real-world graph datasets:** Wiki-CS [A], Polblogs [B], Cora [C], and Chameleon [D] (App. H.2.2)
- We **extended** our analysis of the effect of graph density and homophily **to multiple GNNs** and sample-wise certification (App. H.2.6)
- Next to a transductive setting, we now also investigate an **inductive setting**, where the graph grows at test time (App. H.4)
- **Added** another classification method, namely an SVM with a linear kernel (App. H.3)

Furthermore, we made the following **updates** to the paper:

- We expanded on the **computational complexity** discussion in Lines 270-275
- We clarify the exactness of our method in the abstract (Line 26), and introduction (Line 100), moved and expanded the discussion of **finite-width certification** to the preliminaries in Lines 150-158
- We extended our discussion on finite-width certification by deriving **model-specific bounds** (App. E)
- We added a discussion on the **generality of our certificates** in Lines 523-530
- We added a **detailed technical comparison** to QPCert [E] in App. F


[A] Mernyei and Cangea. Wiki-cs: A wikipedia-based benchmark for graph neural networks. ICML GRL+ Workshop, 2022.

[B] Adamic et al. The political blogosphere and the 2004 US election: divided they blog. International Workshop on Link discovery 2005.

[C] McCallum et al. Automating the construction of internet portals with machine learning. Information Retrieval 2000.

[D] Rozemberczki et al. Multi-scale attributed node embedding. Journal of Complex Networks 2021.

[E] Gosch et al. Provable Robustness of (Graph) Neural Networks Against Data Poisoning and Backdoor Attacks. Arxiv 2024.

**EDIT:** (26.11.) Added model-specific bounds and updated line numbers and App. references.

---

### Meta-Review · Area_Chair_p5p1 · 2024-12-17

**Metareview:**

This paper provides an asymptotically exact robustness certificate for graph neural networks against label flipping attacks through neural tangent kernel (NTK) and mixed-integer formulation. The results are provided on both synthetic and real-world graph benchmarks. After rebuttal discussion, most reviewers concerns are addressed, with only the scalability concern/trade-off of the proposed methods, limiting applicability to small-scale datasets. The authors acknowledge this (current) limitation and discussed the trade-off and tested a strategy to relax the certificate in Appendix A and Appendix H. Overall, this paper provides some interesting ideas and findings and hence acceptance is recommended.

**Additional Comments On Reviewer Discussion:**

The major concerns by reviewers are the following:
1. The proposed robustness certificate is not exact unless the neural network has infinite width, because the authors leverage neural tangent kernel to derive the certificate.
-- In response, the authors acknowledged this point and make it more clear in the revised draft that their certificate is asymptotically exact and hold with high probability when the network width is sufficiently large.

2. The scalability and trade-off of calculating such asymptotically exact robustness certificate
-- The authors acknowledged this point and provided further discussion on the complexity, scalability, and trade-off in the appendix.

3. Real-world examples beyond synthetic data
-- The authors provided additional results on multiple benchmarks to show the applicability of their method

---

### Decision · Program_Chairs · 2025-01-22

Accept (Spotlight)